# DEMOCRATIZING EVALUATION OF DEEP MODEL IN­TERPRETABILITY THROUGH CONSENSUS

## ABSTRACT

Deep learning interpretation tools, such as (Bau et al., 2017; Ribeiro et al., 2016; Smilkov et al., 2017), have been proposed to explain and visualize the ways that deep neural network (DNN) classifiers make predictions. However, the success of these methods highly relies on human subjective interpretations, i.e., the ground truth of interpretations, such as feature importance ranking or locations of visual objects, when evaluating the interpretability of the DNN classifiers on a specific task. For tasks that the ground truth of interpretations is not available, we propose a novel framework *Consensus* incorporating an ensemble of deep models as the committee for interpretability evaluation. Given any task/dataset, *Consensus* first obtains the interpretation results using existing tools, e.g., LIME (Ribeiro et al., 2016), for every model in the committee, then aggregates the results from the entire committee and approximates the "ground truth" of interpretations through voting. With such *quasi-ground-truth*, *Consensus* evaluates the interpretability of a model through matching its interpretation result and the approximated one, and ranks the matching scores together with committee members, so as to pursue the absolute and relative interpretability evaluation results. We carry out extensive ex­periments to validate *Consensus* on various datasets. The results show that *Con­sensus* can precisely identify the interpretability for a wide range of models on ubiquitous datasets that the ground truth is not available. Robustness analyses further demonstrate the advantage of the proposed framework to reach the con­sensus of interpretations through simple voting and evaluate the interpretability of deep models. Through the proposed *Consensus* framework, the interpretability evaluation has been democratized without the need of ground truth as criterion.

## 1 INTRODUCTION

Due to the over-parameterization nature (Allen-Zhu et al., 2019), deep neural networks (DNNs) (Le­Cun et al., 2015) have been widely used to handle machine learning and artificial intelligence tasks, however it is often difficult to understand the prediction results of DNNs despite the very good per­formance. To interpret the DNN classifiers' behaviors, a number of interpretation tools (Bau et al., 2017; Ribeiro et al., 2016; Smilkov et al., 2017; Sundararajan et al., 2017; Zhang et al., 2019; Ahern et al., 2019) have been proposed to recover or visualize the ways that DNNs make decisions.

**Preliminaries.** For example, Network Dissection (Bau et al., 2017) uses a large computer vision dataset with a number of visual concepts identified/localized in every image. Given a convolutional neural network (CNN) model for interpretability evaluation, it recovers the visual features used by the model for the classification of every image via intermediate-layer feature maps, then matches the visual features with the labeled visual concepts to estimate the interpretability of the model as the intersection-over-union (IoU) between the activated feature maps and labeled locations of visual objects. Related tools that interpret CNNs through locating importation subregions of visual features in the feature maps have been proposed in (Zhou et al., 2016; Selvaraju et al., 2020; Chattopadhay et al., 2018; Wang et al., 2020a).

Apart from investigating the inside of complex deep networks, (Ribeiro et al., 2016; van der Linden et al., 2019; Ahern et al., 2019) proposed to use simple linear or tree-based models to surrogate the predictions made by the DNN model over the dataset through local or global approximations, so as to capture the variation of model outputs with the interpolation of inputs in feature spaces. Then,

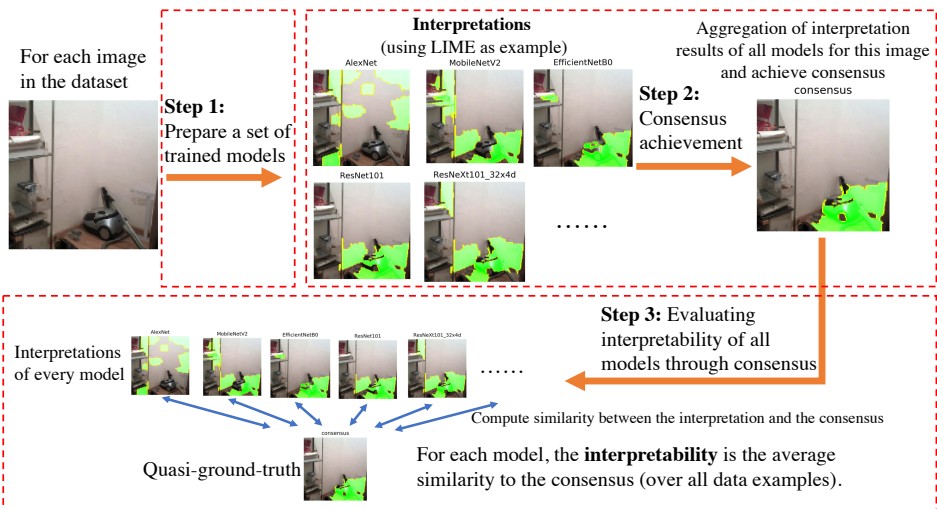

Figure 1: Using *Consensus* to evaluate the interpretability of DNN classifiers with a dataset. For every image in the dataset, *Consensus* (1) prepares set of trained models as committees, (2) aggregates interpretation results from every model to approximate the ground truth of interpretation, and (3) compares the interpretation of every model to the aggregated one to evaluate the interpretability.

with the surrogate model, these methods interpret the DNN model as ways the model uses features for predictions, e.g., ranking of feature importance, and compare the results with the ground truth labeled by human experts to evaluate interpretability. Besides the use of linear interpolation for surrogates, many algorithms, like SmoothGrad (Smilkov et al., 2017), Integrated Gradients (Sundararajan et al., 2017), DeepLIFT (Shrikumar et al., 2017), and PatternNet (Kindermans et al., 2018) have been proposed to estimate the input feature importance as the way to interpret the models, so as to interpret the model predictions by highlighting the importation subregions in the input. In addition to the above methods, (Zhang et al., 2018a; 2019) proposed to learn a graphical model to clarify the process of making decision at a semantic level. Note that obtaining the interpretation of a model is an algorithmic procedure to explain the model (Samek et al., 2017). On the other hand, through the comparing the interpretation results with the human labeled ground truth, the interpretability evaluation aims at estimating the degree to which a human (expert) can consistently predict the model's result (Kim et al., 2016; Doshi-Velez & Kim, 2017).

In summary, the ground truth of interpretation results (usually labeled by human experts) is indispensable to all above methods for interpretability evaluations and comparisons, no matter ways they interpret the models, e.g., visual concepts detecting (Bau et al., 2017), and feature importance ranking for either local (Ribeiro et al., 2016) or global (Ahern et al., 2019) interpretations. While the datasets with visual objects labeled/localized and/or the importance features ranked have offered in some specific areas, the unavailability of ground truths also limits the generalization of these methods to interpret brand new models on the new tasks/datasets ubiquitously. There is thus the need of a method being able to evaluate the interpretability of models on the datasets where the ground truth of interpretation results is not available.

**Our contributions.** In this paper, we study the problem of evaluating the interpretability of DNN classifiers on the datasets without ground truth of interpretation results. The basic idea of *Consensus* is to leverage the interpretability of known models as reference to predict the interpretability of new models on new tasks/datasets. Especially, in terms of general purpose perception tasks, we have already obtained a number of reliable models with decent interpretability, such as ResNets, DenseNets and so on. With a new dataset, one could use interpretation tools (Ribeiro et al., 2016; Smilkov et al., 2017) to obtain the interpretation results of these models, then aggregate interpretation results as the reference. Then for any model, one could evaluate interpretability of the model through comparing its interpretation results with the reference.

Specifically, as illustrated in Figure 1, we propose a novel framework named *Consensus* that uses a large number of known models as a committee for interpretability evaluation. Given any task/dataset, Consensus first obtains the interpretation results for every model in the committee using existing interpretation tools, e.g., LIME (Ribeiro et al., 2016), then aggregates the results from

the entire committee and reaches the consensus of interpretations through voting. With the *quasi-ground-truth*, Consensus evaluates the interpretability of each model through matching its interpretation result and the approximated one, and ranks the matching scores of committee members, so as to pursue the absolute and relative interpretability evaluation results without the ground truth of interpretations labeled by human experts. More specifically, we make contributions as follows.

- We study the problem of interpretability evaluation on the datasets without human labeled ground truth of interpretations. To the best of our knowledge, this work is the first to study the problem of evaluating the interpretability of DNNs while the ground truths of interpretation results are not available, by addressing the technical issues of voting and committees.

- We design and implement *Consensus*, a novel interpretability evaluation framework that incorporates a wide range of alternating interpretation tools, such as LIME (Ribeiro et al., 2016), SmoothGrad (Smilkov et al., 2017), to interpret the model as the variation of outputs over the interpolations of inputs (in feature spaces), from massive perspectives (e.g., local or global interpretation, tree-based or linear surrogate and so on), and carries out the interpretability evaluation through the voting based on interpretation results of models in the committee.

- We carry out extensive experiments to validate Consensus on a wide range of models on new ubiquitous tasks/datasets that the ground truth is not available, and exploit the quantifiable metric of model interpretability to report the overall interpretability evaluation results (Section 3). Case studies (Section 4) confirm the effectiveness of Consensus and show the closeness of Consensus based results to the ground truth of interpretations. Robustness analyses (Section 5) further demonstrate the advantage of the committee that the factors including the use of basic interpretation algorithms, the types of networks in the committee, and the size of committee would have few affects on the Consensus based interpretability evaluation results.

## 2 CONSENSUS: A FRAMEWORK OF INTERPRETABILITY EVALUATION

In this section, we introduce our proposed framework, namely *Consensus*, which incorporates existing interpretations, such as LIME (Ribeiro et al., 2016) and SmoothGrad (Smilkov et al., 2017), to enable DNN interpretability evaluation without the use of human labeled interpretation results as reference/ground truth. Specifically, *Consensus* generalizes a simple *electoral system*, and consists of three steps: (1) Committee Formation with Deep Models, (2) Committee Voting for Consensus Achievement[1], and (3) Consensus-based Interpretability Evaluation, as follow.

**Committee Formation with Deep Models.** Given a number of deep neural networks (which are well-known with decent performance on common perception tasks) and a target task with a dataset (potentially without ground truth of interpretation results), *Consensus* first trains the given neural networks (from scratch or fine-tuned) using the dataset. Then, *Consensus* forms the post-trained networks as a committee of models, noted as $\mathcal{M}$, and considers the variety of interpretability of models in the committee that would establish the references for interpretability comparisons and evaluation. Note that while our research assumes the human labeled ground truth of interpretation results is not available in the given task/dataset for interpretability evaluation, the labels of samples are requested when handling classifications and regression tasks.

**Committee Voting for Consensus Achievement.** With the committee of trained models and the target task/dataset for interpretation, *Consensus* first leverages an existing interpretation tool $\mathcal{A}$, e.g., $\mathcal{A}$ can be LIME (Ribeiro et al., 2016) or SmoothGrad (Smilkov et al., 2017) alternatively, to obtain the interpretation results of every model in the committee on every sample in the dataset. Given some sample $d_i$, we note the obtained interpretation results of all models as $\boldsymbol{L}$. Then, *Consensus* proposes a voting procedure that aggregates $\boldsymbol{L}$ to achieve the consensus $\boldsymbol{c}$ as the *quasi-ground-truth* of the interpretation for the sample $d_i$. Specifically, $\boldsymbol{c}_k = \frac{1}{m} \sum_{i=1}^{m} \frac{\boldsymbol{L}_{ik}^2}{\|\boldsymbol{L}_i\|}$ for LIME and $\boldsymbol{c}_k = \frac{1}{m} \sum_{i=1}^{m} \frac{\boldsymbol{L}_{ik} - min(\boldsymbol{L}_i)}{max(\boldsymbol{L}_i) - min(\boldsymbol{L}_i)}$ for SmoothGrad. In summary, *Consensus* adopts a normalization-averaging procedure to obtain the quasi-ground-truth of interpretations for the sample. To the end, the consensus has been achieved through obtaining the collections of quasi-ground-truth for every sample in the target dataset based on committee voting.

**Consensus-based Interpretability Evaluation.** Given the quasi-ground-truths as the consensus of the whole committee, the proposed algorithm evaluates the interpretability of every model in the

---

[1]We are not intending to connect our work with the multi-agent research though we use the term "consensus achievement".

committee by considering the similarity between the interpretation result of each individual model and the consensus of the whole committee. Specifically, for the interpretations and consensus based on LIME, *Consensus* uses cosine similarity between the flattened vectors of interpretation of each model and the consensus. Then *Consensus* quantifies the interpretability of the model through the mean of similarity measures over all samples. For the results based on SmoothGrad (visual feature importance in pixel levels), *Consensus* follows a similar procedure, where the proposed algorithm uses Radial Basis Function ($exp(-\frac{1}{2}(||\boldsymbol{a} - \boldsymbol{b}||/\sigma)^2)$) for the similarity measurement. We rank all models in the committee using their similarities to the consensus and consider the top/bottom models with good/bad interpretability in the committee.

---

**Algorithm 1:** *Consensus* Framework Pseudocode. The functions interpret(), aggregate() and sim() are described in the main text and detained in Algorithm 2 in Appendix E.

---

1   function **Consensus**($\mathcal{D}, \mathcal{A}$)
   **Input**  : A dataset $\mathcal{D}$ containing $n$ examples $\{d_i\}_{i=1,\cdots,n}$ and an interpretation algorithm $\mathcal{A}$.
   **Output:** $\boldsymbol{s} \in \mathbb{R}^m$, where each element $\boldsymbol{s}_j$ indicates the interpretability of each model $\boldsymbol{M}_j$ in $\mathcal{M}$.
   `/* Step 1:  Committee Formation with Deep Models` $\mathcal{M}$      `*/`
2   Prepare $\mathcal{M}$ containing $m$ models $\{\boldsymbol{M}_j\}_{j=1,\cdots,m}$, i.e., the committee of deep models.
3   $\boldsymbol{S} = zeros(n, m)$ `// Initialize an empty` $n \times m$ `matrix for storing the`
     `interpretability scores of` $m$ `models on` $n$ `data sample.`
4   **for** $i$ *in* $1, \cdots, n$ **do**
5      $\boldsymbol{L} = zeros(m, p_i)$
6      **for** $j$ *in* $1, \cdots, m$ **do**
7         $\boldsymbol{L}_j = $ interpret$(\mathcal{A}, \boldsymbol{M}_j, d_i)$
8      **end**
     `/* Step 2:  Committee Voting for Consensus Achievement at` $d_i$    `*/`
9      $\boldsymbol{c} = $ aggregate$(\boldsymbol{L})$ `//` $\boldsymbol{c} \in \mathbb{R}^{p_i}$`, consensus as quasi-ground-truth`
     `/* Step 3:  Consensus-based Interpretability Evaluation at` $d_i$    `*/`
10     **for** $j$ *in* $1, \cdots, m$ **do**
11       $\boldsymbol{S}_{ij} = $ sim$(\boldsymbol{L}_j, \boldsymbol{c})$ `// the score of` $\boldsymbol{M}_j$ `at` $d_i$
12     **end**
13   **end**
14   **for** $j$ *in* $1, \cdots, m$ **do**
15     $\boldsymbol{s}_j = average(\boldsymbol{S}_{\cdot j})$ `// average score for each model over` $n$ `samples`
16   **end**
17   return $\boldsymbol{s}$

---

These three steps of *Consensus* are illustrated in Figure 1 and formalized in Algorithm 1. Note that for any new models for interpretability evaluation, *Consensus* includes them as members of committee together with a number of known models, and performs above procedures to obtain their interpretability evaluation results (absolute evaluation results) as well as the ranking in the committee (relative evaluation results). In this way, one can clearly position the interpretability and potentials (in terms of performance) of the new models on the new tasks among the known models, even when the ground truth of interpretation results are not available.

## 3   OVERALL EVALUATION AND RESULTS

In this section, we use the image classification as the target task for interpretation and interpretability evaluation of deep models. We first introduce the settings of image classification tasks in details, as the setups of our experiments. Then, we present the overall results of *Consensus*, where we could observe its capacity of evaluating the interpretability of models, with connections to the model performance, while the ground truth of interpretations is not used. Through the comparisons with interpretability evaluation based on LIME and SmoothGrad algorithms using human labeled ground truth, the effectiveness of *Consensus* has been evaluated.

### 3.1   EVALUATION SETUPS

Here we present the setups of our experiments from following perspectives.

**Datasets and Models.** For overall evaluation and comparisons, we use two image classification datasets ImageNet (Deng et al., 2009) for ubiquitous visual objects and CUB-200-2011 (Welinder et al., 2010) for birds respectively. Note that ImageNet provides the class label for every image; CUB-200-2011 dataset includes the class label and pixel-level segmentation for the bird in every

image, where the pixel annotations of visual objects have been considered as the ground truth of interpretations (Bau et al., 2017). In this way, we evaluate interpretability of models using *Consensus*, and compare the results with the existing algorithms based on the ground truth.

For the fair comparisons, we use more than 80 models publicly available from PaddlePaddle[2], which have been trained using ImageNet dataset. We also derive models based on CUB-200-2011 dataset through standard fine-tuning procedures. In our experiments, we include these models in two committees based on ImageNet and CUB-200-2011 respectively for the interpretability evaluation.

**Baselines.** We consider two interpretation algorithms, LIME (Ribeiro et al., 2016) and Smooth-Grad (Smilkov et al., 2017). Specifically, LIME surrogates the interpretation as the assignment of visual feature importance to superpixels (Vedaldi & Soatto, 2008), SmoothGrad outputs the interpretation results as the visual feature importance over pixels. In this way, we can evaluate the flexibility of the proposed *Consensus* framework over interpretation results from diverse sources (i.e., linear surrogates vs. input gradients) and in multiple granularity (i.e., feature importance in superpixel/pixel-levels). Note that both algorithms use mean Average Precision (mAP) between their interpretation results and the ground truth (i.e., pixel level segmentation, if available) as the measure of interpretability evaluation.

**Metrics.** Given the similarity measures between the model's interpretation results and consensus of the committee, the proposed algorithm is used to rank the interpretability of every model in the committee. In this way, we compare the ranking list of model interpretability based on *Consensus* evaluation results with the ranking list based on the ground truth, so as to understand how well the proposed *Consensus* framework can approximate the evaluation results of model interpretability without the use of ground truth. More specifically, visual comparisons, Pearson correlation coefficients (which characterize the linearity between two variables, e.g., model performance vs. interpretability evaluation) and significance tests have been used as the metrics for overall evaluation and comparisons.

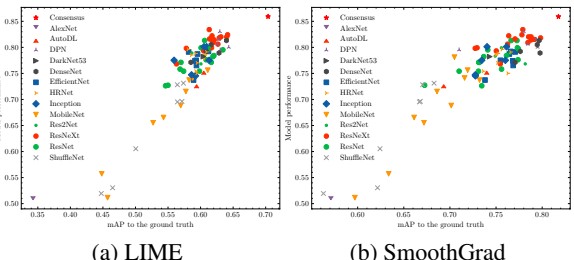

(a) LIME        (b) SmoothGrad

Figure 2: Correlation between the Model Performance (Testing Accuracy) and the Ground Truth based Interpretability Evaluation using (a) LIME and (b) SmoothGrad with CUB-200-2011 over 85 models. Pearson correlation coefficients are 0.927 (with p-value 4e-37) for LIME and 0.916 (with p-value 9e-35) for SmoothGrad. The points "Consensus" here refer to the testing accuracy of the ensemble of networks in the committee by probabilities averaging and voting (in y-axis), as well as the mAP between the Consensus results and the ground truth (in x-axis).

Note that we do not have the ground truth of interpretations for ImageNet. However, it is still possible for us to access the interpretability evaluation through connecting *Consensus* outputs with the generalization performance of the models. Figure 2 illustrates an example of correlations between interpretability evaluation results based on ground truth and the testing accuracy of 85 models over CUB-200-2011 dataset, where we can see strong, significant, and consistent correlations between model performance and the interpretability.

### 3.2 OVERALL COMPARISONS

In Figure 3, we present the interpretability evaluation results using CUB-200-2011, and the overall comparisons between *Consensus* and the ground truth based, where we plot the scatter points (i.e., *Consensus* results in x-axis vs. ground truth based results in y-axis) of the two comparisons using LIME and SmoothGrad respectively. In both comparisons, *Consensus* performs almost-identically as the one based on ground truth with strong Spearman's correlation (which characterizes the consistency between two ranking lists) with significance tests passed.

To enable the similar comparisons using ImageNet dataset (where the ground truth of interpretations is not available), we connect the *Consensus* results with the model performance (testing accuracy) of models, as the model performance and the ground truth based interpretability evaluation results are usually correlated (please see also in Figure 2 and section 3.1). In Figure 3, we present the

---

[2]https://github.com/PaddlePaddle/models/blob/release/1.8/PaddleCV/
image_classification/README_en.md#supported-models-and-performances

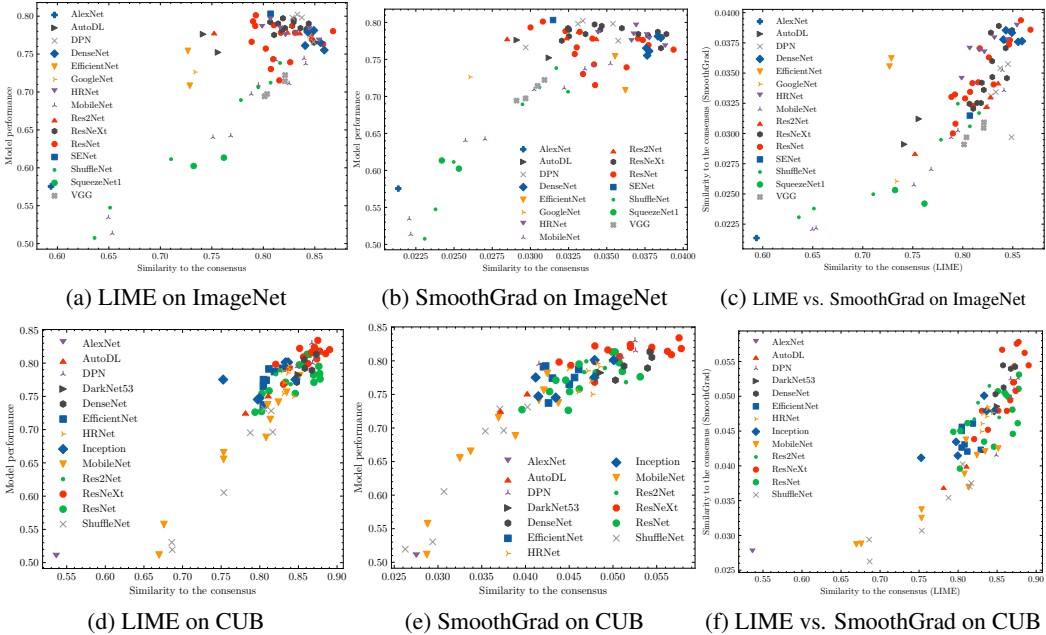

(a) LIME on ImageNet     (b) SmoothGrad on ImageNet     (c) LIME vs. SmoothGrad on ImageNet

(d) LIME on CUB     (e) SmoothGrad on CUB     (f) LIME vs. SmoothGrad on CUB

Figure 4: Model performance v.s. similarity to the consensus using LIME (a,d) and SmoothGrad (b,e) over 81 models on ImageNet (a,b) and 85 models on CUB-200-2011 (d,e). The third column shows the similarity to the consensus of SmoothGrad interpretations v.s. similarity to the consensus of LIME interpretations on ImageNet committee (c) and CUB-200-2011 committee (f). Pearson correlation coefficients are (a) 0.8087, (b) 0.783, (c) 0.825, (d) 0.908, (e) 0.880 and (f) 0.854.

correlations between the *Consensus* results (without the use of ground truth) and the model performance for both LIME and SmoothGrad using ImageNet and CUB-200-2011 datasets. Specifically, in Figure 4 (a-b) and (d-e), we present the comparisons between model performance (in y-axis) and the *Consensus* results (in x-axis) using LIME (a,d) and SmoothGrad (b,e) on ImageNet (a,b) and CUB-200-2011 (d,e) respectively.

All correlations here are strong with significance tests passed, though in some local areas of the correlation plots between model performance and interpretability evaluation the trends are not always consistent. It has been observed that some extremely large networks work well with the datasets, while they are lack of interpretability (Bau et al., 2017). In this way, we could conclude that, in an overall manner, interpretability evaluation results based on *Consensus* using both LIME and SmoothGrad over the two datasets are correlated to model performance with significance.

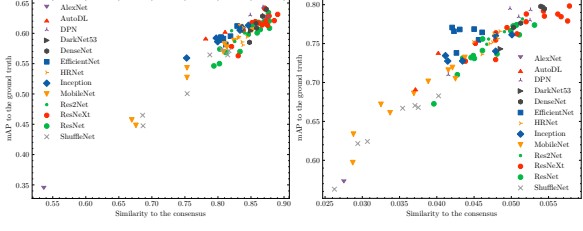

(a) Using LIME     (b) Using SmoothGrad

Figure 3: Ground-Truth-based v.s. *Consensus*-based Interpretability Evaluation using (a) LIME and (b) SmoothGrad with CUB-200-2011 Datasets over 85 models (the Committee). Spearman's correlation coefficients are 0.885 (with p-value 3e-29) for LIME and 0.906 (with p-value 8e-33) for SmoothGrad.

## 3.3 COMPARISON RESULTS WITH NETWORK DISSECTION (BAU ET AL., 2017)

Here we compare the results of *Consensus* with ground truth based interpretability evaluation solution – Network Dissection (Bau et al., 2017). With ImageNet dataset, Network Dissection gave a ranking list of models (w.r.t the model interpretability) as follows: ResNet152 > DenseNet161 > VGG16 > GoogLeNet > AlexNet. We report two ranking lists based on *Consensus* with detailed numbers for every architecture above, which has been demonstrated in Figure 4 (a, LIME): DenseNet161 (0.849) ≈ ResNet152 (0.846) > VGG16 (0.821) > GoogLeNet (0.734) > AlexNet (0.594); and (b, SmoothGrad): DenseNet161 (0.038) ≈ ResNet152 (0.037) > VGG16 (0.030) >

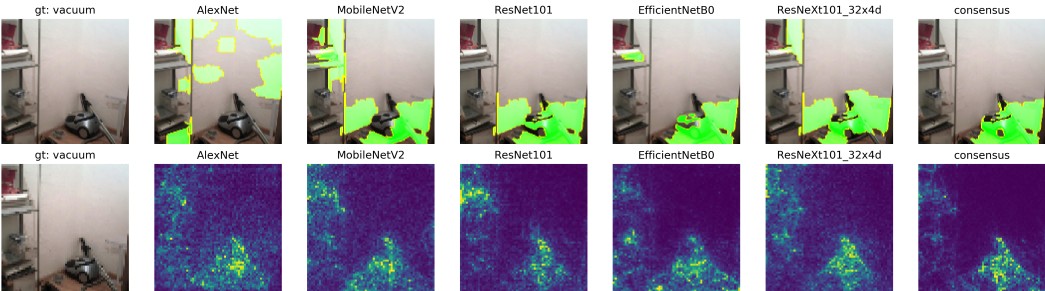

Figure 5: Visual comparisons between consensus and the interpretation results of CNNs using LIME (in the upper line) and SmoothGrad (in the lower line) based on an image from ImageNet, where the ground truth of interpretation results is not available.

GoogLeNet (0.026) > AlexNet (0.021). The three ranking lists are almost identical, except the comparisons between DenseNet161 and ResNet152, where in the both lists of *Consensus*, DenseNet161 is similar to ResNet152 with marginally elevated interpretability, while Network Dissection considers ResNet152 is more interpretable than DenseNet161.

We believe the results from *Consensus* and Network Dissection are close enough from the perspectives of ranking lists, and the difference may be caused by the different ways that *Consensus* and Network Dissection evaluate the interpretability. *Consensus* matches the interpretations with the visual objects on images (due to the results of LIME and SmoothGrad), while Network Dissection counts the number of neurons activated by the visual objects and patterns (color, materials, textures, scenes, and parts). Furthermore, Network Dissection evaluates the interpretability of deep models using the Broden dataset with densely labeled visual objects and patterns (Bau et al., 2017), while *Consensus* does not need additional dataset or ground truths of interpretations. In this way, the results by *Consensus* and Network Dissection might be slightly different.

## 4 CASE STUDIES

In this section, we discuss several technical issues on *Consensus* for ground-truth-free evaluation of DNN interpretability, using a set of case studies.

**Qualification and Effectiveness of Committee-based Voting**   In our research, *Consensus* proposes to replace the ground truth with a committee of networks, and it trusts the voting results as the interpretations of ground truth. Thus, we have to verify (1) whether the (consensus achieved by the) committee would approximate to the ground truth, and (2) whether voting is an effective way to express the interpretation results of the whole committee.

To achieve the goal, we consider the committee as an ensemble of networks, and it classifies data via committee-based voting through averaging the probability outputs of the member networks. In Figure 2, we compare the committee (as an ensemble of networks, entitled with "Consensus") with other architectures for the evaluation of model performance (testing accuracy) and the interpretability evaluation with the ground truth (i.e., mAP between consensus achieved by the committee and the ground truth), using both LIME and SmoothGrad over CUB-200-2011 dataset. The comparison results show that the committee is of the best testing accuracy with significant advantages compared to other models, while the committee is also with the highest mAP between its consensus interpretations and the ground truth compared to other models. In this way, we can confirm the qualification of the committee, as well as the effectiveness of the committee-based voting.

**Closeness of Consensus to the Ground Truth**   In addition to the mAP measurement (illustrated in Figure 2) between consensus and the ground truth on CUB-200-2011 dataset, we also visualize the examples to compare the consensus achieved by the committee, interpretation results of individual networks using both LIME and SmoothGrad, and (optionally) the ground truth labeled for both ImageNet and CUB-200-2011 datasets in Figures 5 and 6 respectively. The comparison shows that the consensus can clearly segment the visual objects related to the classification from the background of images, and it would be closer to the ground truth (if available) than the individual networks. Both quantitative results in Figure 2 and the visual comparisons in Figures 5 and 6 validate the closeness of consensus to the ground truth of interpretations.

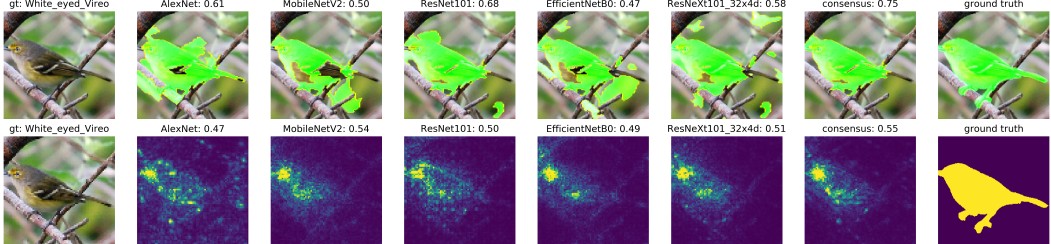

Figure 6: Visual comparisons between consensus and the interpretation results of CNNs using LIME (in the upper line) and SmoothGrad (in the lower line) based on an image from CUB-200-2011, where the ground truth of interpretation results is available as pixel-wise annotations and the mean Average Precision (mAP) are measured for interpretability evaluation.

## 5 ROBUSTNESS ANALYSES

In this section, we discuss several factors, including the use of basic interpretation algorithms (e.g., LIME and SmoothGrad), the size of committee, and the candidate pool for models in the committee, that would affect the proposed *Consensus* framework.

**Consistency of LIME and SmoothGrad**    While *Consensus* adopts LIME and SmoothGrad as the basic interpretation algorithms, the interpretation results from these two algorithms are not exactly the same. Even though the granularity of interpretation results are different, which causes mismatching in mAP estimation with ground truth, the interpretability evaluation results of *Consensus* based on the two algorithms are generally consistent. The consistency has been confirmed by Figure 4 (c, f), where the overall results of Consensus based on LIME is strongly correlated to SmoothGrad over all models. This shows that the proposed *Consensus* framework can work well with a wide spectrum of basic interpretation algorithms.

**Consistency of Cross-Committee Interpretability Evaluation**    In real-word applications, the committee-based evaluation makes the results inconsistent in a committee-by-committee manner. In this work, we are interested in whether the interpretability evaluation is consistent against the change of committee (e.g., using different sets of models). Given 16 ResNet models as the targets, we form 20 independent committees through combining the 16 models with 10–20 models randomly drawn from the networks presented in Figure 4. In each of these 20 independent committees, we use *Consensus* to evaluate the interpretability of 16 ResNet models and rank them accordingly. We then estimate the Pearson correlation coefficients between any of these 20 ranking lists and the list in Figure 4 (a), where the mean correlation coefficient is 0.96 with the standard deviation 0.04. Thus, we can say the interpretability evaluation based on randomly picked committees would be consistent.

**Convergence over Committee Sizes**    To understand effects of committee sizes to interpretability evaluation, we run *Consensus* using committees of various sizes formed with networks randomly picked up from the pools. In Figure 7, we plot and compare the performance of the consensus with increasing committee sizes, where we estimate the mAP between the ground truth and the consensus based on the random committees of different sizes and 20 random trials have been done for every single size

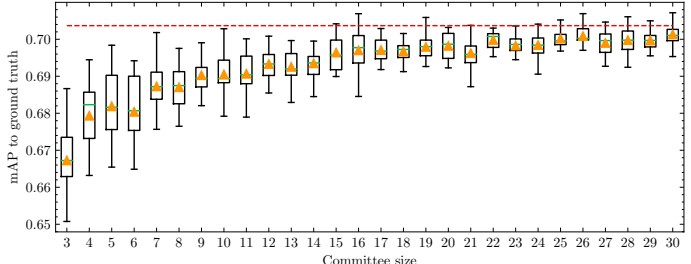

Figure 7: Convergence of mAP between the ground truth and the consensus results based on committees of increasing sizes, using LIME on CUB-200-2011. The green lines are the mean values and the orange triangles are the median value. The red dashed line is the mAP of the consensus based on the complete committee of the original 85 models.

independently. It shows that the curve of mAP would quickly converge to the complete the committee, while the consensus based on a small proportion of committee (e.g., 15 networks) works good enough even compared to the complete committee with 85 networks.

**Applicability with Random Committees over Other Datasets**    To demonstrate the applicability of *Consensus* with varying committees over other datasets, we continue our experiments using net-

works randomly picked up from the pool on other datasets, including Stanford Cars 196 (Krause et al., 2013), Oxford Flowers 102 (Nilsback & Zisserman, 2008) and Foods 101 (Bossard et al., 2014) in Figure 8, where we consider the connections between the interpretability and model performance (e.g., the testing accuracy, inspired by Figure 2). The results confirm that, when the ground truth of interpretations is not available, our framework is still capable of identifying the interpretability for a wide range of models on ubiquitous datasets/tasks.

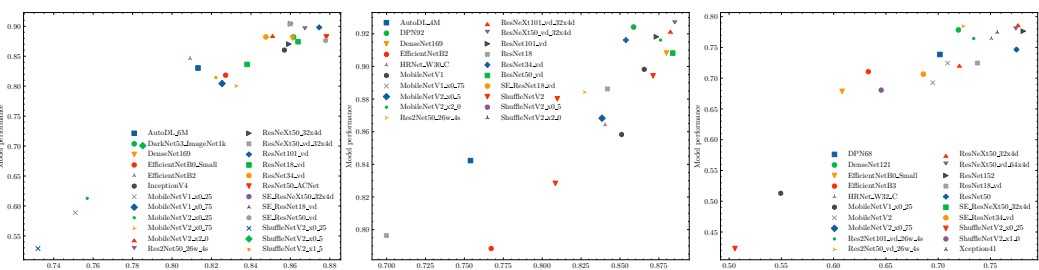

Figure 8: Model performance versus Results of Consensus using LIME on Stanford Cars 196 (Krause et al., 2013), Oxford Flowers 102 (Nilsback & Zisserman, 2008) and Foods 101 (Bossard et al., 2014). Pearson correlation coefficients are 0.9522, 0.8785 and 0.9134 respectively.

## 6    CONCLUSION AND FUTURE WORK

We have proposed a novel framework *Consensus* for evaluating the interpretability of deep models while avoiding the use of ground truth for interpretations. Specifically, *Consensus* forms a committee of deep models and generalizes an electoral system to reach the consensus of interpretation results using some basic interpretation algorithms via committee-based voting. Then, for every model in the committee, *Consensus* computes the similarity between its interpretation results and the aggregated consensus, then it ranks the models in the committee accordingly, so as to pursue the absolute score (i.e., similarities to the consensus) and relative results (the rank in the committee) for interpretability evaluation. To validate *Consensus*, we carry out extensive experimental studies using 85 deep models including ResNets, DenseNets, and so on, on top of 5 datasets, including ImageNet, CUB-200-2011 and so on, in comparison with interpretation algorithms of three categories, including LIME (Ribeiro et al., 2016), SmoothGrad (Smilkov et al., 2017), and Network Dissection (Bau et al., 2017). The results show that (1) *Consensus* can evaluate the interpretability of models on the datasets even when the ground truth of interpretation results are not available, (2) the consensus of interpretation results aggregated from the committee could well approximate the ground truth of interpretations, (3) the interpretability evaluation results delivered by *Consensus* correlates to the model performance (testing accuracy) strongly and significantly, (4) the factors including the use of basic interpretation algorithms, the types of networks in the committee, and the size of committee would not affect the results of interpretability evaluation with *Consensus*.

**Discussion.** For the interpretability evaluation of any models on any dataset, the ground truth based evaluation approaches rely on human subjective interpretations, while *Consensus* can automate this process by just training a few more models and approximating the ground truth of interpretations. We thus conclude that the interpretability evaluation can be democratized through an electoral system constructed by the DNN models themselves, rather than the use of human labeled ground truth as criterion. We discuss the future work in three different directions. (1) In terms of methodologies, the proposed *Consensus* framework considers the segmentation of visual features as interpretations for vision tasks and adopts simple voting mechanism to aggregate results from the committee of models. We believe that the contributions made in this work are complementary with visual objects segmentation and voting. The use of advanced segmentation (Chen et al., 2017a; He et al., 2017) and ensemble learning methods (Dietterich, 2000; Hinton et al., 2015) would further improve the proposed framework. (2) In terms of applications, following the steps of *Consensus*, on medical or financial domains where interpretations for black-box models are urged, the quasi-ground-truth of interpretations and the interpretability evaluation of models could be easily obtained. (3) Instead of using trained models of different architectures as committee members, models of common or even the same architectures trained using various training strategies would also form an interesting committee for analyzing the interpretability of models based on different training strategies.

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

## A MORE VISUALIZATION RESULTS

We present more visualization results of Consensus, where the samples are from ImageNet and CUB-200-2011.

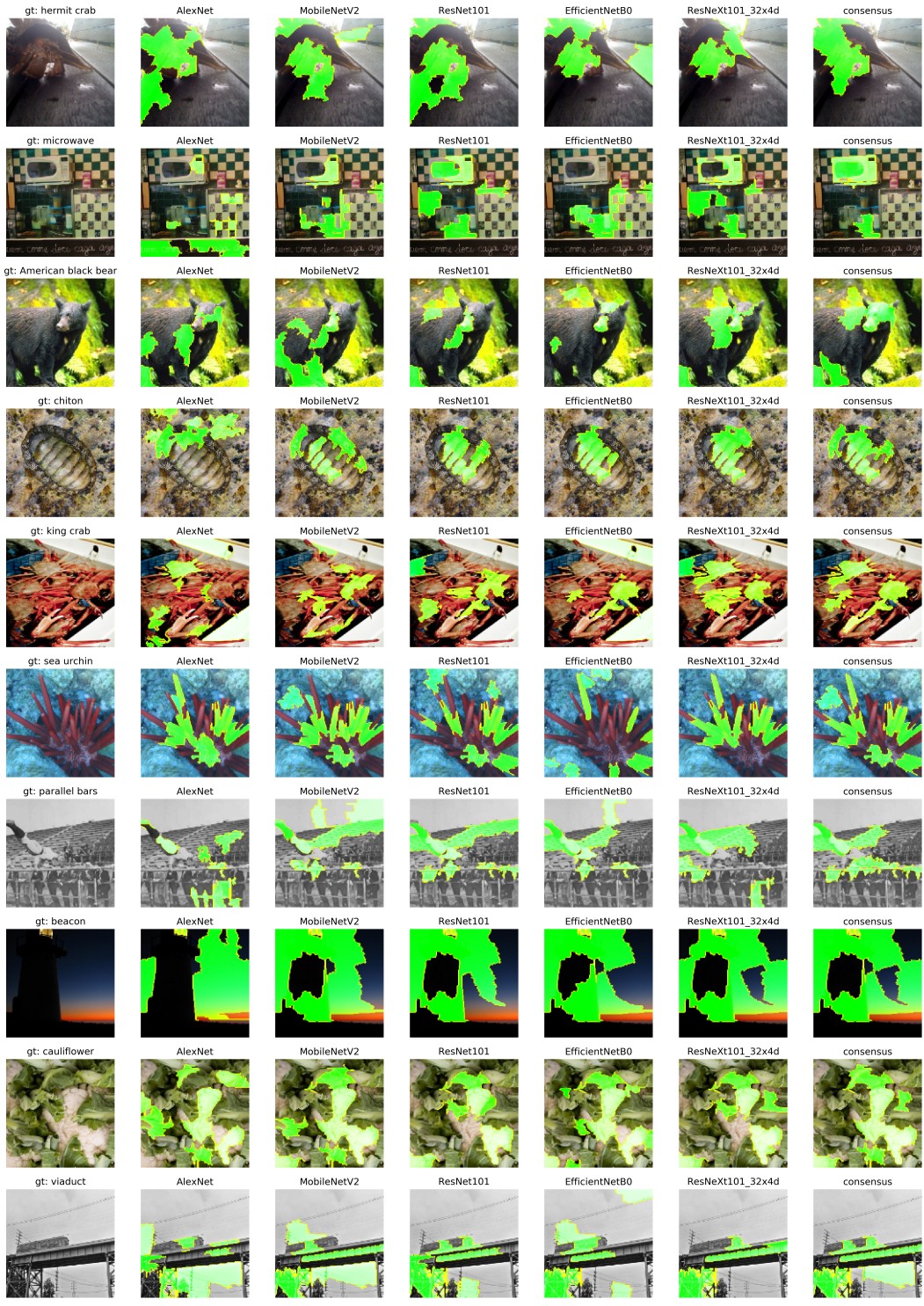

Figure 9: More visual comparisons between the consensus interpretation and the interpretation results of CNNs with LIME on samples from ImageNet, where the ground truth of interpretation results is not available. Note that consensus is the *Consensus* aggregated interpretation.

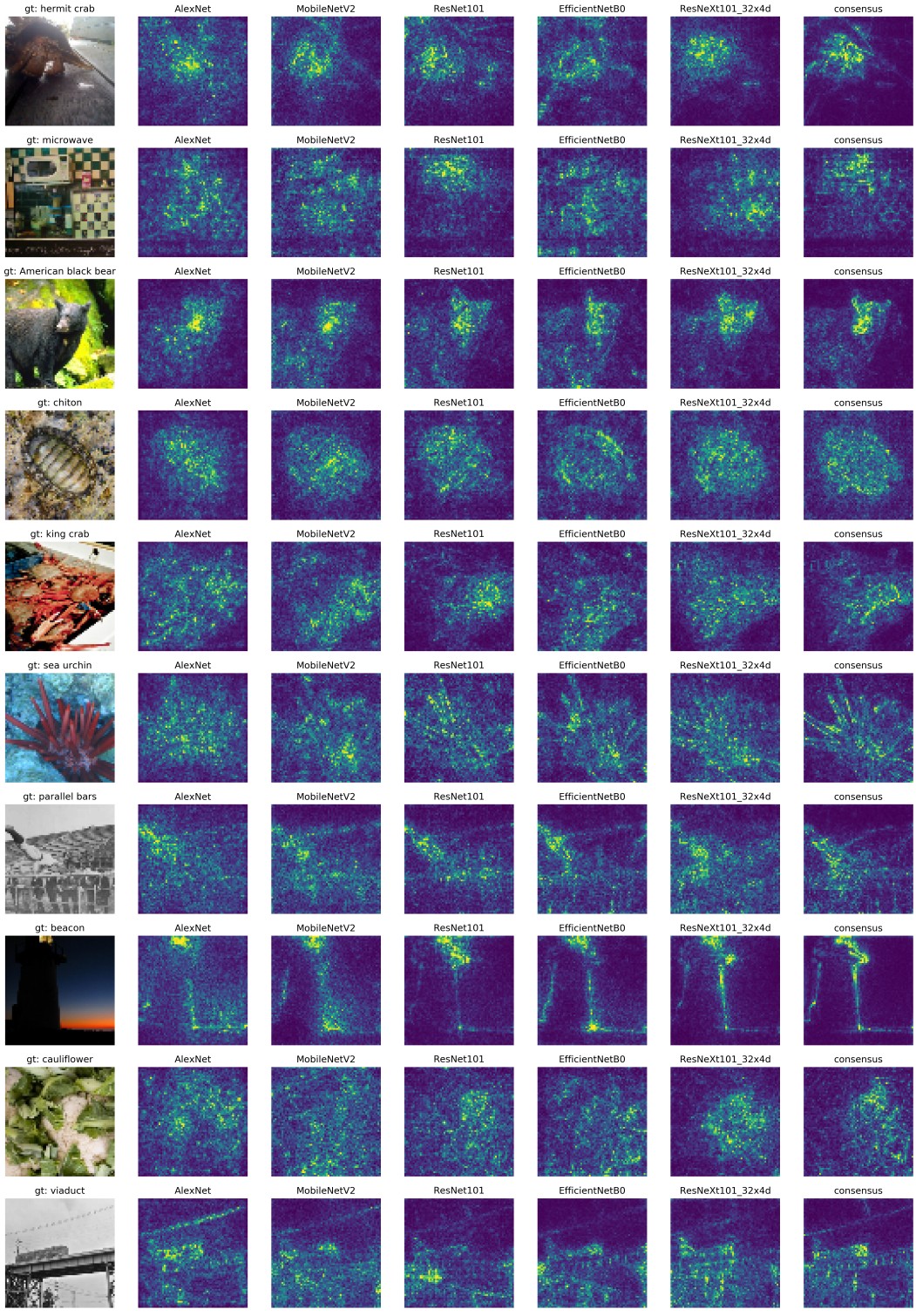

Figure 10: More visual comparisons between the consensus interpretation and the interpretation results of CNNs with SmoothGrad on samples from ImageNet, where the ground truth of interpretation results is not available. Note that consensus is the *Consensus* aggregated interpretation.

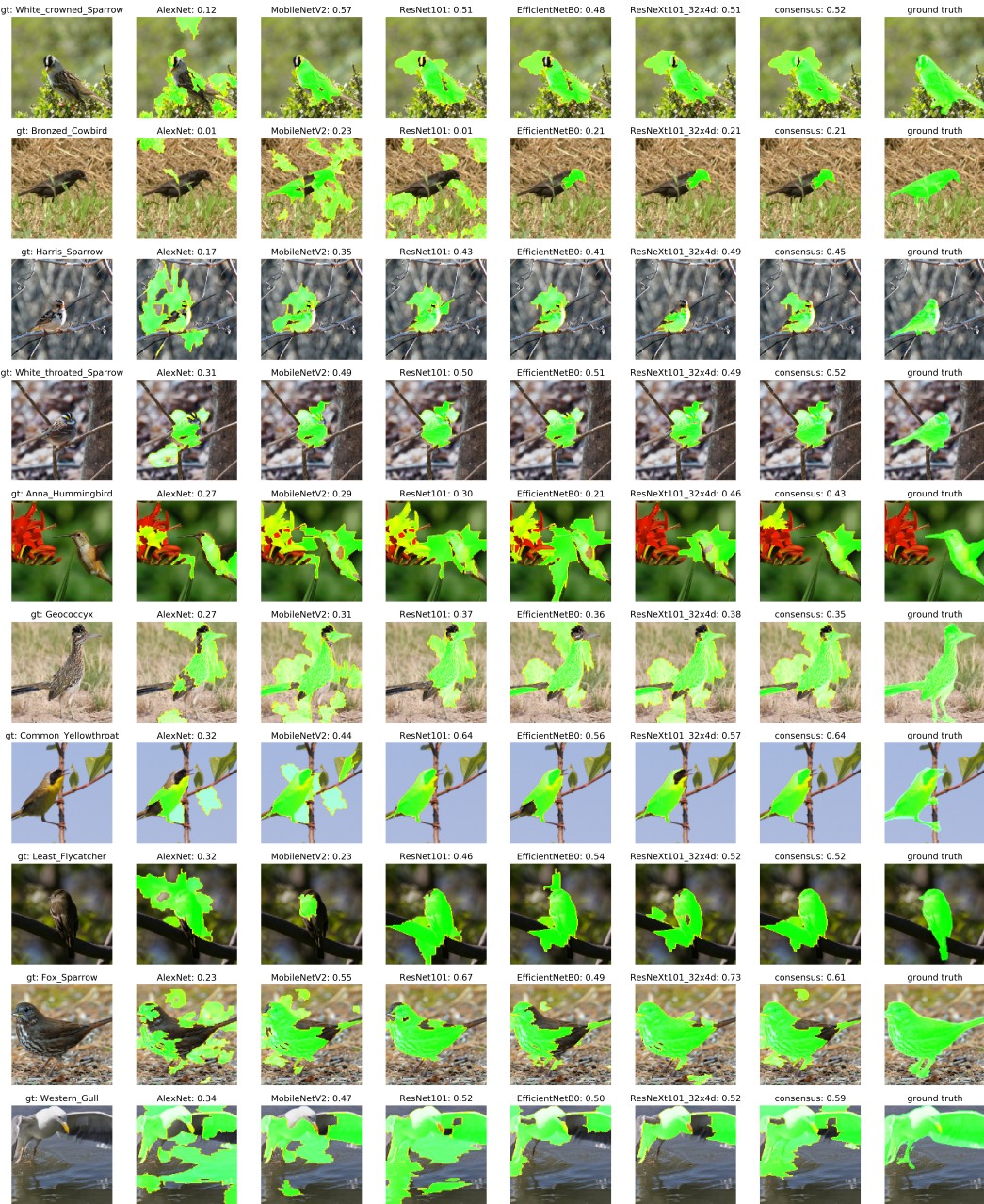

Figure 11: More visual comparisons between the consensus interpretation and the interpretation results of CNNs with LIME on samples from CUB-200-2011, where the ground truth of interpretation results is available as pixel-wise annotations and the mAPs are measured for interpretability evaluation. Note that consensus is the *Consensus* aggregated interpretation.

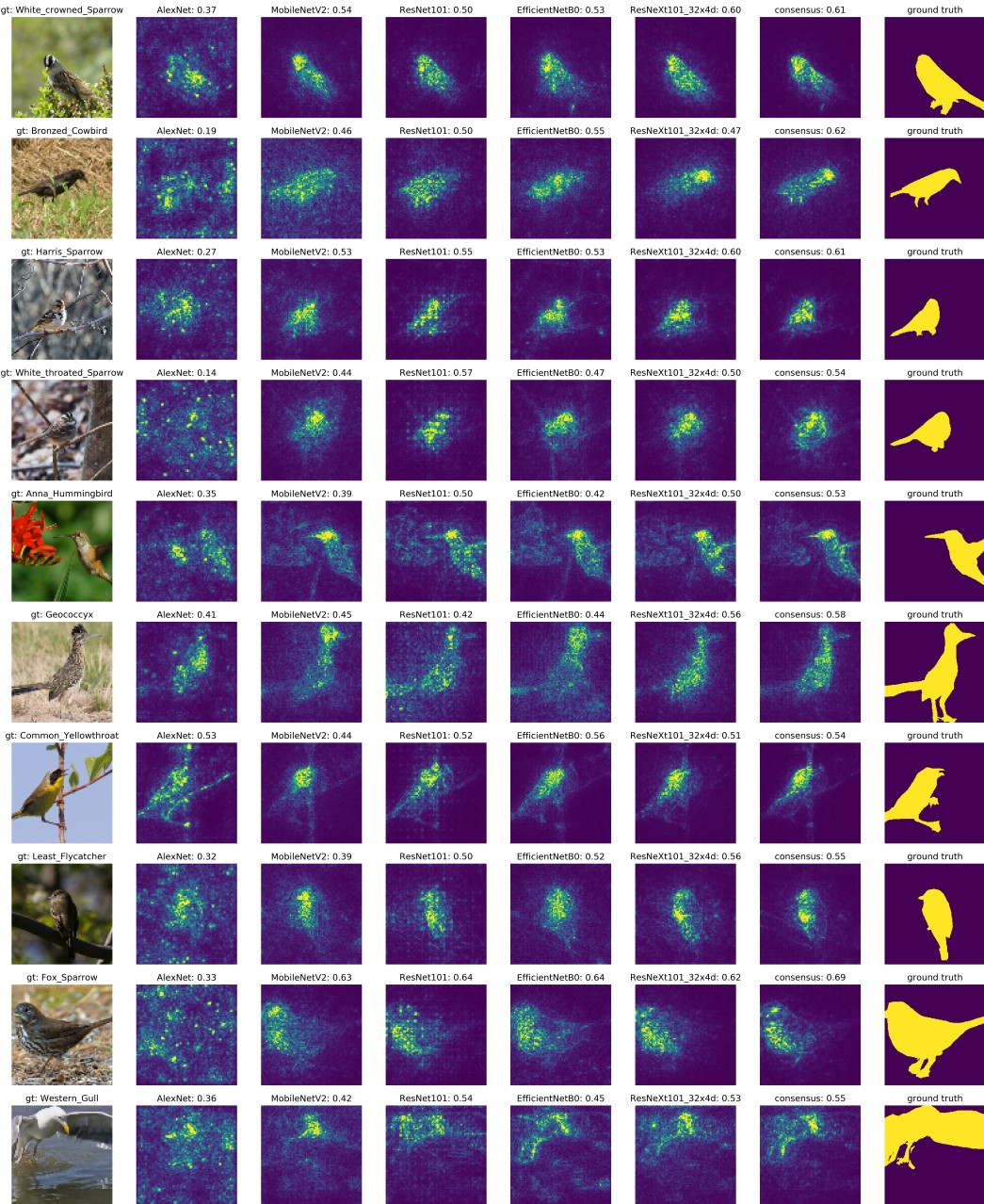

Figure 12: More visual comparisons between the consensus interpretation and the interpretation results of CNNs with SmoothGrad on samples from CUB-200-2011, where the ground truth of interpretation results is available as pixel-wise annotations and the mAPs are measured for interpretability evaluation. Note that consensus is the *Consensus* aggregated interpretation.

## B    EXPERIMENTAL DETAILS

We present the technique details for the experiments in the main text.

### B.1    COMMITTEE FORMATIONS

There are around 100 publicly available deep models trained on ImageNet from PaddlePaddle[3]. We first exclude some very large models that take much more computation resources. Then for the consistency of computing superpixels, we include only the models that take images of size $224 \times 224$ as input, resulting 81 models for the committee based on ImageNet. Since there are already a large number of available models, we choose to not include more models by aligning the superpixels in different sizes of images.

As for CUB-200-2011 (Welinder et al., 2010), similarly we first exclude the very large models. Then we follow the standard procedures (Sermanet et al., 2014; Simonyan & Zisserman, 2015) for fine-tuning ImageNet-pretrained models on CUB-200-2011. For simplicity, we use the same training setup for all pre-trained models (learning rate 0.01, batch size 64, SGD optimizer with momentum 0.9, resize to 256 being the short edge, randomly cropping images to the size of $224 \times 224$), and obtain 85 models that are well trained. Different hyper-parameters may help to improve the performance of some specific networks, but for the same reason of the large number of available models, we choose to not search for better hyper-parameter settings.

Given the convergence over committee sizes (Figure 7), which shows that a committee of more than 15 models works good enough, thus we randomly choose around 20 models for Stanford Cars 196 (Krause et al., 2013), Oxford Flowers 102 (Nilsback & Zisserman, 2008) and Foods 101 (Bossard et al., 2014), following the same training procedure as CUB-200-2011.

### B.2    INTERPRETATION ALGORITHMS

To interpret a model, LIME (Ribeiro et al., 2016) on vision tasks first performs a superpixel segmentation (Vedaldi & Soatto, 2008) for an image, then generates samples by randomly masking some superpixels and computing the outputs through the model, and finally fits the model outputs with the set of superpixels as input by a linear regression model. The linear weights then presents the feature importance in the superpixel level as the interpretation result.

The gradients of model output w.r.t. input can partly identify influential pixels, but due to the saturation of activation functions in the deep networks, the vanilla gradient is usually noisy. SmoothGrad (Smilkov et al., 2017) reduces the visual noise by repeatedly adding small random noises to the input so as to get a list of corresponding gradients, which are averaged for the final interpretation result.

## C    RESNET FAMILY

We show the zoomed plot of ResNet family (whose name contains "ResNet" key word) in the ImageNet-LIME committee of 81 models in Figure 13 (a). Meanwhile, we also present the results using ResNet family as committee in Figure 13 (b). These two subfigures have no large difference, which further confirms the consistency of ranking models in different committees.

## D    REFERENCES OF NETWORK STRUCTURES

Many structures of deep neural networks have been evaluated in this paper, including AlexNet (Krizhevsky et al., 2012), ResNet (He et al., 2016), ResNeXt (Xie et al., 2017), SEResNet (Hu et al., 2018), ShuffleNet (Zhang et al., 2018b; Ma et al., 2018), MobileNet (Howard et al., 2017; Sandler et al., 2018; Howard et al., 2019), VGG (Simonyan & Zisserman, 2015), GoogleNet (Szegedy et al., 2015), Inception (Szegedy et al., 2015), Xception (Chollet, 2017), DarkNet (Redmon et al., 2016; Redmon & Farhadi, 2018), DenseNet (Huang et al., 2017), DPN (Chen et al., 2017b), SqueezeNet

---

[3]`https://github.com/PaddlePaddle/models/blob/release/1.8/PaddleCV/`
`image_classification/README_en.md#supported-models-and-performances`

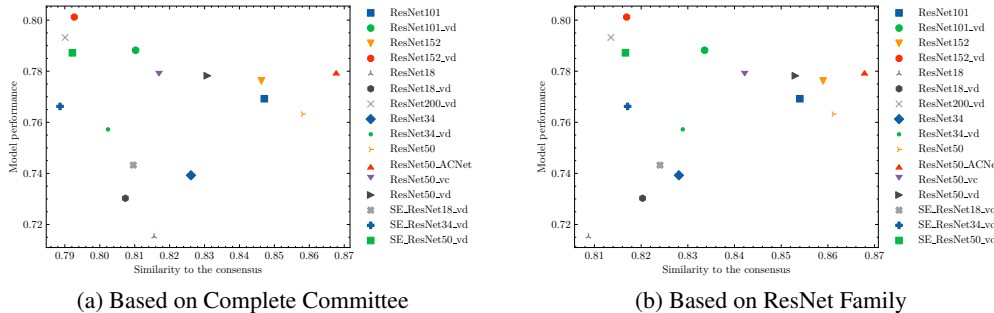

(a) Based on Complete Committee        (b) Based on ResNet Family

Figure 13: Model performance v.s. similarity to the consensus of LIME on ResNet family. The consensus of (a) is voted by the complete committee on ImageNet (81 models), while the consensus of (b) is voted by ResNet family (16 models).

(Iandola et al., 2016), EfficientNet (Tan & Le, 2019), Res2Net (Gao et al., 2019), HRNet (Wang et al., 2020b), Darts (Liu et al., 2018), AcNet (Ding et al., 2019) and so on.

## E    COMPLETE PSEUDOCODE OF *Consensus*

Figure 1 shows an illustrative pipeline of our framework Consensus of evaluating the interpretability of models without the need of the ground truth of interpretation, and Algorithm 2 gives a complete process of Consensus in pseudocode.

## F    NUMERICAL REPORT OF MAIN PLOTS

Due to the large number of deep models evaluated, Figure 2, 3 and 4 grouped those that are of the same architecture. Here, we report all of the corresponding numerical results in Table 1.

## G    VISUALIZATION OF COCO IMAGES

For further showing the effectiveness of Consensus, we visualize several random images from MS-COCO (Lin et al., 2014), shown in Figure 14.

---

**Algorithm 2:** *Consensus* Framework Pseudocode

---

**1** function **sim**($\boldsymbol{a}, \boldsymbol{b}$)

   **Input** : Two vectors or tensors.

   **Output:** a scalar as the similarity between $\boldsymbol{a}$ and $\boldsymbol{b}$ with appropriate normalization approaches.

   /* This function uses cosine similarity for LIME interpretations and Radial Basis Function, $exp(-\frac{1}{2}(||\boldsymbol{a} - \boldsymbol{b}||/\sigma)^2)$. */

**2**

**3** function **aggregate**($\boldsymbol{L}$)

   **Input** : $\boldsymbol{L}$, a collection of interpretations of $m$ models for one given data sample.

   **Output:** $\boldsymbol{c}$, the consensus among $m$ model for the interpretation of the given data sample.

   /* This function returns the *quasi-ground-truth* of the interpretation for the given sample, basically it is equivalent to a normalization-averaging procedure in this paper. Specifically, $\boldsymbol{c}_k = \frac{1}{m}\frac{\sum_{i=1}^{m} \boldsymbol{L}_{ik}^2}{||\boldsymbol{L}_i||}$ for LIME, and $\boldsymbol{c}_k = \frac{1}{m}\sum_{i=1}^{m} \frac{\boldsymbol{L}_{ik} - min(\boldsymbol{L}_i)}{max(\boldsymbol{L}_i) - min(\boldsymbol{L}_i)}$ for SmoothGrad. */

**4**

**5** function **interpret**($\mathcal{A}, \boldsymbol{M}, d$)

   **Input** : An interpretation algorithm $\mathcal{A}$, a trained model $\boldsymbol{M}$ and a data sample $d \in \mathbb{R}^p$, where $\mathbb{R}^p$ is the feature domain and the dimension $p$ may vary with the sample $d$.

   **Output:** $\boldsymbol{\alpha} \in \mathbb{R}^p$, where the elements indicate the importances of input features.

   /* This function is an implementation of a typical interpretation algorithm like LIME (number of superpixels as $p$), SmoothGrad (number of pixels as $p$) or others. */

**6**

**7** function **Consensus**($\mathcal{D}, \mathcal{A}$)

   **Input** : A dataset $\mathcal{D}$ containing $n$ examples $\{d_i\}_{i=1,\cdots,n}$ and an interpretation algorithm $\mathcal{A}$.

   **Output:** $\boldsymbol{s} \in \mathbb{R}^m$, where each element $\boldsymbol{s}_j$ indicates the interpretability of each model $\boldsymbol{M}_j$ in $\mathcal{M}$.

   /* Step 1: Committee Formation with Deep Models $\mathcal{M}$ */

**8** Prepare $\mathcal{M}$ containing $m$ models $\{\boldsymbol{M}_j\}_{j=1,\cdots,m}$, i.e., the committee of deep models.

**9** $\boldsymbol{S} = zeros(n, m)$ // Initialize an empty $n \times m$ matrix for storing the interpretability scores of $m$ models on $n$ data sample.

**10** **for** $i$ *in* $1, \cdots, n$ **do**

**11**    $\boldsymbol{L} = zeros(m, p_i)$

**12**    **for** $j$ *in* $1, \cdots, m$ **do**

**13**       $\boldsymbol{L}_j$ = interpret($\mathcal{A}, \boldsymbol{M}_j, d_i$)

**14**    **end**

      /* Step 2: Committee Voting for Consensus Achievement at $d_i$ */

**15**    $\boldsymbol{c}$ = aggregate($\boldsymbol{L}$) // $\boldsymbol{c} \in \mathbb{R}^{p_i}$, consensus as quasi-ground-truth

      /* Step 3: Consensus-based Interpretability Evaluation at $d_i$ */

**16**    **for** $j$ *in* $1, \cdots, m$ **do**

**17**       $\boldsymbol{S}_{ij}$ = sim($\boldsymbol{L}_j, \boldsymbol{c}$) // the score of $\boldsymbol{M}_j$ at $d_i$

**18**    **end**

**19** **end**

**20** **for** $j$ *in* $1, \cdots, m$ **do**

**21**    $\boldsymbol{s}_j = average(\boldsymbol{S}_{\cdot j})$ // average score for each model over $n$ samples

**22** **end**

**23** return $\boldsymbol{s}$

---

Table 1: Numerical report of model performance and similarity to the consensus using LIME and SmoothGrad over 81 models on ImageNet in sub-table(a) corresponding to Figure 4 (a, b, c), and over 85 models on CUB-200-2011 in sub-table(b) corresponding to Figure 2, 3 and 4 (d, e, f).

(a) on ImageNet

| | perf. | interpretability scores of Consensus (LIME) | interpretability scores of Consensus (SmoothGrad) |
|---|---|---|---|
| AlexNet | 0.575 | 0.594 | 0.0214 |
| AutoDL_4M | 0.752 | 0.756 | 0.0312 |
| AutoDL_6M | 0.776 | 0.741 | 0.0291 |
| DPN107 | 0.798 | 0.828 | 0.0331 |
| DPN131 | 0.802 | 0.833 | 0.0334 |
| DPN68 | 0.766 | 0.849 | 0.0297 |
| DPN92 | 0.775 | 0.845 | 0.0357 |
| DPN98 | 0.798 | 0.837 | 0.0354 |
| DenseNet121 | 0.755 | 0.859 | 0.0376 |
| DenseNet161 | 0.781 | 0.849 | 0.0383 |
| DenseNet169 | 0.765 | 0.855 | 0.0376 |
| DenseNet201 | 0.779 | 0.843 | 0.0385 |
| DenseNet264 | 0.761 | 0.841 | 0.0378 |
| EfficientNetB0 | 0.754 | 0.727 | 0.0355 |
| EfficientNetB0_Small | 0.708 | 0.729 | 0.0362 |
| GoogleNet | 0.726 | 0.734 | 0.0260 |
| HRNet_W18_C | 0.766 | 0.854 | 0.0388 |
| HRNet_W30_C | 0.776 | 0.832 | 0.0378 |
| HRNet_W32_C | 0.781 | 0.845 | 0.0376 |
| HRNet_W40_C | 0.773 | 0.822 | 0.0366 |
| HRNet_W44_C | 0.782 | 0.817 | 0.0368 |
| HRNet_W48_C | 0.794 | 0.807 | 0.0369 |
| HRNet_W64_C | 0.784 | 0.799 | 0.0344 |
| MobileNetV1 | 0.711 | 0.825 | 0.0322 |
| MobileNetV1_x0_25 | 0.513 | 0.653 | 0.0222 |
| MobileNetV1_x0_5 | 0.640 | 0.751 | 0.0257 |
| MobileNetV1_x0_75 | 0.697 | 0.788 | 0.0297 |
| MobileNetV2 | 0.742 | 0.812 | 0.0342 |
| MobileNetV2_x0_25 | 0.534 | 0.650 | 0.0221 |
| MobileNetV2_x0_5 | 0.642 | 0.768 | 0.0270 |
| MobileNetV2_x0_75 | 0.709 | 0.795 | 0.0302 |
| MobileNetV2_x1_5 | 0.737 | 0.841 | 0.0336 |
| MobileNetV2_x2_0 | 0.744 | 0.840 | 0.0352 |
| Res2Net101_vd_26w_4s | 0.780 | 0.752 | 0.0285 |
| Res2Net50_14w_8s | 0.781 | 0.823 | 0.0324 |
| Res2Net50_26w_4s | 0.780 | 0.835 | 0.0343 |
| Res2Net50_vd_26w_4s | 0.790 | 0.828 | 0.0332 |
| ResNeXt101_32x4d | 0.784 | 0.843 | 0.0371 |
| ResNeXt101_vd_32x4d | 0.795 | 0.830 | 0.0347 |
| ResNeXt101_vd_64x4d | 0.784 | 0.821 | 0.0336 |
| ResNeXt152_32x4d | 0.782 | 0.842 | 0.0377 |
| ResNeXt152_64x4d | 0.787 | 0.828 | 0.0383 |
| ResNeXt152_vd_32x4d | 0.792 | 0.807 | 0.0325 |
| ResNeXt152_vd_64x4d | 0.790 | 0.814 | 0.0325 |
| ResNeXt50_32x4d | 0.765 | 0.849 | 0.0385 |
| ResNeXt50_64x4d | 0.784 | 0.836 | 0.0389 |
| ResNeXt50_vd_32x4d | 0.790 | 0.844 | 0.0346 |
| ResNeXt50_vd_64x4d | 0.792 | 0.829 | 0.0360 |
| ResNet101 | 0.769 | 0.847 | 0.0377 |
| ResNet101_vd | 0.788 | 0.810 | 0.0323 |
| ResNet152 | 0.776 | 0.846 | 0.0374 |
| ResNet152_vd | 0.801 | 0.793 | 0.0308 |
| ResNet18 | 0.715 | 0.816 | 0.0342 |
| ResNet18_vd | 0.730 | 0.807 | 0.0334 |
| ResNet200_vd | 0.793 | 0.790 | 0.0300 |
| ResNet34 | 0.739 | 0.826 | 0.0363 |
| ResNet34_vd | 0.757 | 0.802 | 0.0329 |
| ResNet50 | 0.763 | 0.858 | 0.0394 |
| ResNet50_ACNet | 0.780 | 0.868 | 0.0386 |
| ResNet50_vc | 0.778 | 0.817 | 0.0370 |
| ResNet50_vd | 0.778 | 0.831 | 0.0341 |
| SENet154_vd | 0.803 | 0.807 | 0.0315 |
| SE_ResNeXt101_32x4d | 0.781 | 0.818 | 0.0325 |
| SE_ResNeXt50_32x4d | 0.775 | 0.810 | 0.0321 |
| SE_ResNeXt50_vd_32x4d | 0.797 | 0.819 | 0.0342 |
| SE_ResNet18_vd | 0.743 | 0.810 | 0.0342 |
| SE_ResNet34_vd | 0.766 | 0.789 | 0.0330 |
| SE_ResNet50_vd | 0.787 | 0.792 | 0.0332 |
| ShuffleNetV2 | 0.706 | 0.795 | 0.0325 |
| ShuffleNetV2_x0_25 | 0.507 | 0.636 | 0.0231 |
| ShuffleNetV2_x0_33 | 0.547 | 0.651 | 0.0238 |
| ShuffleNetV2_x0_5 | 0.611 | 0.710 | 0.0250 |
| ShuffleNetV2_x1_0 | 0.689 | 0.778 | 0.0295 |
| ShuffleNetV2_x1_5 | 0.712 | 0.807 | 0.0306 |
| ShuffleNetV2_x2_0 | 0.738 | 0.816 | 0.0317 |
| SqueezeNet1_0 | 0.602 | 0.732 | 0.0253 |
| SqueezeNet1_1 | 0.613 | 0.762 | 0.0242 |
| VGG11 | 0.694 | 0.801 | 0.0291 |
| VGG13 | 0.697 | 0.804 | 0.0297 |
| VGG16 | 0.714 | 0.821 | 0.0305 |
| VGG19 | 0.722 | 0.821 | 0.0309 |

(b) on CUB-200-2011

| | perf | interpretability scores of Consensus (LIME) | interpretability scores of Consensus (SmoothGrad) | mAP between g.t. of segmentation and LIME interpretation | mAP between g.t. of segmentation and SmoothGrad interpretation |
|---|---|---|---|---|---|
| AlexNet | 0.507 | 0.536 | 0.0275 | 0.343 | 0.571 |
| AutoDL_4M | 0.728 | 0.781 | 0.0371 | 0.594 | 0.693 |
| AutoDL_6M | 0.754 | 0.811 | 0.0402 | 0.605 | 0.740 |
| DPN107 | 0.830 | 0.867 | 0.0525 | 0.630 | 0.780 |
| DPN131 | 0.800 | 0.868 | 0.0498 | 0.643 | 0.795 |
| DPN68 | 0.795 | 0.849 | 0.0415 | 0.630 | 0.710 |
| DPN92 | 0.806 | 0.872 | 0.0510 | 0.626 | 0.784 |
| DPN98 | 0.815 | 0.877 | 0.0526 | 0.628 | 0.793 |
| DarkNet53_ImageNet1k | 0.782 | 0.850 | 0.0485 | 0.604 | 0.743 |
| DenseNet121 | 0.771 | 0.848 | 0.0503 | 0.585 | 0.771 |
| DenseNet161 | 0.813 | 0.873 | 0.0542 | 0.640 | 0.797 |
| DenseNet169 | 0.792 | 0.858 | 0.0513 | 0.609 | 0.776 |
| DenseNet201 | 0.805 | 0.858 | 0.0544 | 0.616 | 0.795 |
| DenseNet264 | 0.789 | 0.868 | 0.0540 | 0.628 | 0.798 |
| EfficientNetB0 | 0.765 | 0.805 | 0.0450 | 0.594 | 0.769 |
| EfficientNetB0_Small | 0.737 | 0.805 | 0.0426 | 0.589 | 0.738 |
| EfficientNetB1 | 0.775 | 0.805 | 0.0456 | 0.593 | 0.755 |
| EfficientNetB2 | 0.787 | 0.819 | 0.0461 | 0.595 | 0.764 |
| EfficientNetB3 | 0.791 | 0.812 | 0.0421 | 0.582 | 0.771 |
| EfficientNetB4 | 0.792 | 0.829 | 0.0423 | 0.612 | 0.766 |
| EfficientNetB5 | 0.774 | 0.808 | 0.0431 | 0.591 | 0.768 |
| HRNet_W18_C | 0.754 | 0.831 | 0.0461 | 0.592 | 0.736 |
| HRNet_W30_C | 0.770 | 0.832 | 0.0475 | 0.595 | 0.752 |
| HRNet_W32_C | 0.785 | 0.836 | 0.0471 | 0.586 | 0.750 |
| HRNet_W40_C | 0.750 | 0.844 | 0.0476 | 0.594 | 0.763 |
| HRNet_W44_C | 0.788 | 0.830 | 0.0449 | 0.592 | 0.752 |
| HRNet_W48_C | 0.796 | 0.838 | 0.0482 | 0.581 | 0.757 |
| HRNet_W64_C | 0.791 | 0.838 | 0.0485 | 0.609 | 0.766 |
| InceptionV4 | 0.745 | 0.797 | 0.0435 | 0.592 | 0.728 |
| MobileNetV1 | 0.741 | 0.824 | 0.0415 | 0.588 | 0.716 |
| MobileNetV1_x0_25 | 0.557 | 0.676 | 0.0288 | 0.448 | 0.634 |
| MobileNetV1_x0_5 | 0.655 | 0.753 | 0.0325 | 0.527 | 0.672 |
| MobileNetV1_x0_75 | 0.688 | 0.808 | 0.0388 | 0.569 | 0.701 |
| MobileNetV2 | 0.737 | 0.810 | 0.0438 | 0.582 | 0.732 |
| MobileNetV2_x0_25 | 0.511 | 0.670 | 0.0287 | 0.457 | 0.597 |
| MobileNetV2_x0_5 | 0.665 | 0.753 | 0.0337 | 0.543 | 0.661 |
| MobileNetV2_x0_75 | 0.715 | 0.814 | 0.0369 | 0.577 | 0.686 |
| MobileNetV2_x1_5 | 0.756 | 0.835 | 0.0421 | 0.611 | 0.719 |
| MobileNetV2_x2_0 | 0.781 | 0.851 | 0.0425 | 0.605 | 0.705 |
| Res2Net101_vd_26w_4s | 0.799 | 0.853 | 0.0470 | 0.613 | 0.756 |
| Res2Net50_14w_8s | 0.789 | 0.826 | 0.0491 | 0.587 | 0.765 |
| Res2Net50_26w_4s | 0.768 | 0.840 | 0.0515 | 0.601 | 0.782 |
| Res2Net50_vd_26w_4s | 0.783 | 0.821 | 0.0467 | 0.604 | 0.749 |
| ResNeXt101_32x4d | 0.818 | 0.877 | 0.0578 | 0.629 | 0.798 |
| ResNeXt101_32x8d_wsl | 0.768 | 0.831 | 0.0479 | 0.563 | 0.755 |
| ResNeXt101_vd_32x4d | 0.816 | 0.867 | 0.0494 | 0.614 | 0.771 |
| ResNeXt101_vd_64x4d | 0.824 | 0.871 | 0.0520 | 0.642 | 0.778 |
| ResNeXt152_32x4d | 0.815 | 0.872 | 0.0543 | 0.619 | 0.792 |
| ResNeXt152_64x4d | 0.834 | 0.875 | 0.0576 | 0.613 | 0.779 |
| ResNeXt152_vd_32x4d | 0.820 | 0.872 | 0.0520 | 0.640 | 0.788 |
| ResNeXt152_vd_64x4d | 0.822 | 0.852 | 0.0479 | 0.618 | 0.764 |
| ResNeXt50_32x4d | 0.809 | 0.856 | 0.0567 | 0.619 | 0.785 |
| ResNeXt50_64x4d | 0.814 | 0.885 | 0.0562 | 0.621 | 0.788 |
| ResNeXt50_vd_32x4d | 0.806 | 0.874 | 0.0508 | 0.627 | 0.762 |
| ResNeXt50_vd_64x4d | 0.820 | 0.890 | 0.0544 | 0.631 | 0.785 |
| ResNet101 | 0.784 | 0.878 | 0.0511 | 0.620 | 0.761 |
| ResNet101_vd | 0.813 | 0.864 | 0.0499 | 0.606 | 0.766 |
| ResNet152 | 0.799 | 0.859 | 0.0506 | 0.601 | 0.773 |
| ResNet152_vd | 0.797 | 0.851 | 0.0507 | 0.613 | 0.774 |
| ResNet18 | 0.726 | 0.794 | 0.0449 | 0.546 | 0.735 |
| ResNet18_vd | 0.754 | 0.846 | 0.0428 | 0.598 | 0.710 |
| ResNet200_vd | 0.813 | 0.861 | 0.0502 | 0.618 | 0.773 |
| ResNet34 | 0.758 | 0.812 | 0.0461 | 0.569 | 0.756 |
| ResNet34_vd | 0.771 | 0.833 | 0.0435 | 0.570 | 0.731 |
| ResNet50 | 0.776 | 0.878 | 0.0531 | 0.609 | 0.774 |
| ResNet50_ACNet | 0.782 | 0.870 | 0.0481 | 0.619 | 0.737 |
| ResNet50_vd | 0.795 | 0.876 | 0.0461 | 0.634 | 0.741 |
| SE_ResNeXt101_32x4d | 0.793 | 0.838 | 0.0452 | 0.605 | 0.750 |
| SE_ResNeXt50_32x4d | 0.798 | 0.821 | 0.0438 | 0.578 | 0.727 |
| SE_ResNeXt50_vd_32x4d | 0.799 | 0.863 | 0.0479 | 0.617 | 0.729 |
| SE_ResNet18_vd | 0.727 | 0.802 | 0.0396 | 0.550 | 0.673 |
| SE_ResNet34_vd | 0.754 | 0.803 | 0.0450 | 0.574 | 0.731 |
| SE_ResNet50_vd | 0.771 | 0.870 | 0.0446 | 0.616 | 0.732 |
| ShuffleNetV2 | 0.696 | 0.817 | 0.0375 | 0.571 | 0.668 |
| ShuffleNetV2_x0_25 | 0.519 | 0.687 | 0.0263 | 0.448 | 0.563 |
| ShuffleNetV2_x0_33 | 0.530 | 0.686 | 0.0294 | 0.465 | 0.622 |
| ShuffleNetV2_x0_5 | 0.605 | 0.753 | 0.0307 | 0.500 | 0.624 |
| ShuffleNetV2_x1_0 | 0.695 | 0.788 | 0.0354 | 0.564 | 0.667 |
| ShuffleNetV2_x1_5 | 0.728 | 0.815 | 0.0371 | 0.564 | 0.670 |
| ShuffleNetV2_x2_0 | 0.731 | 0.806 | 0.0402 | 0.574 | 0.683 |
| Xception41 | 0.801 | 0.833 | 0.0501 | 0.605 | 0.761 |
| Xception41_deeplab | 0.775 | 0.753 | 0.0412 | 0.559 | 0.734 |
| Xception65 | 0.801 | 0.837 | 0.0479 | 0.609 | 0.740 |
| Xception65_deeplab | 0.747 | 0.800 | 0.0415 | 0.586 | 0.728 |
| Xception71 | 0.775 | 0.846 | 0.0479 | 0.613 | 0.760 |
| Consensus | 0.859 | N/A | N/A | 0.704 | 0.818 |

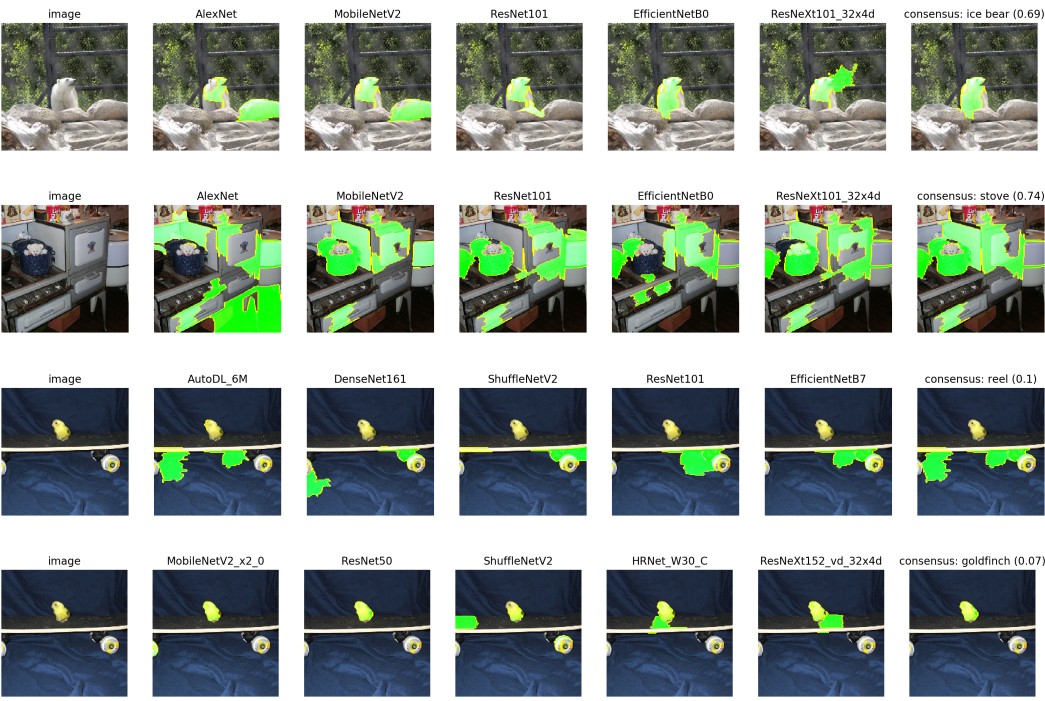

Figure 14: Visualization of images from the MS-COCO dataset (Lin et al., 2014) for showing the effectiveness of our framework Consensus, where the predicted label with probability is noted. Note that consensus is the *Consensus* aggregated interpretation.

