# OpenReview forum: "Democratizing Evaluation of Deep Model Interpretability through Consensus"
_ICLR.cc/2021/Conference — Reject_

### Official Review · AnonReviewer4 · 2020-10-27
**Limited technical contribution, not very clearly explained.**

**Rating:** 3
**Confidence:** 3

**Review:**

The authors propose a novel method, called Consensus, using an ensemble of deep learning architectures, to study the interpretability of models, when ground truth of interpretations is not available.

The high-level idea of the paper is promising, i.e. evaluating interpretability without the need for human-labelled groundtruth, which is tedious and expensive to collect, and for this task, can be subjective.
The proposed method consists of three stages: forming a committee of deep models, aggregating the results in “quasi ground-truth”, and ranking models of the committee based on the similarity with the quasi ground-truth.
However, it is not clear from the paper how the “committee voting” is performed (is it averaging, majority voting), and what does the method bring in addition to just an ensemble of multiple CNN architectures.

Great points of the paper:
- using pretrained models to overcome the lack of ground-truth (which would be subjective and costly to collect).
- comparison of the consensus results with several architectures, and consensus vs ground truth (via mAP).

Points on which the paper can be improved:
1. Clarity of description of the method
- there is no clear formal definition of how the committee voting works; adding formulas would improve readability.
- no explanation on how RBF is used to address the dimensionality differences when comparing scores;
- it is difficult to understand from the paper how the numbers (interpretability scores) are obtained (Sec. 3.3.); please add equations and reference them in the text - especially for correlation score and aggregating scores.
- please clarify how the significance tests were performed
- Since the figures are cluttered, it might be worth considering reporting the per CNN scores in a table (few models in the main paper) and the rest in the appendix.

2. Experimental comparisons with other methods
- the proposed method compares on object domain only, ImageNet and CUB200-2001 (birds). The method compares with Network Dissection (Bau et al, 2017), but no results on Broden dataset are reported. This dataset is particularly interesting, as it is unifying several datasets, spread across multiple domains (objects, scenes, objet parts, textures and materials).
- Why is it necessary to have the models trained from scratch? (Sec. 2, committee). What would change if the models would be just finetuned?
- What does the current method bring in addition to LIME and SmoothGrad aggregated over a large number of architectures?
Datasets as MS-COCO / LVIS provide ground truth labels for object segmentation, which are more diverse than CUB / ImageNet.

3. Style / visuals:
- axis labels and text on the plots are too small;  pick a smaller number of models, the figures are too cluttered. The full comparison could be in tables, in appendix / supplementary material.
- “significance tests” instead of “significant tests”.
- are the 4 decimals in the Pearson coefficient necessary? (Fig. 1 caption)
- please report the exact p-value.
- Fig. 5 - please overlay the contour of ground truth over the other visualizations.

The overall idea is simple - ensemble of CNN architectures, to generate quasi-ground-truth for interpretability. Not clear what "interpretability" is -- and how this would be different from object segmentation masks.
The paper would greatly benefit from a more formal explanation of how the models are scored, and how the consensus is computed.
It would add value to emphasize more how the proposed method could leverage existing trained models for datasets on which such labels are non-existent;
It would also prove the effectiveness of the method, if at least one more datapoint would be reported, on datasets with object labels (e.g. MS-COCO / LVIS).

---

> ### Author Response · Authors · 2020-11-18
> **Response to ICLR 2021 Conference Paper1036 AnonReviewer4 (Part 1/2)**
>
> Many thanks for your review and the constructive comments. We do appreciate your efforts in helping us to improve the manuscript. Actually, we have uploaded a modified version with some updates in appendix. We will update the formal revised version of the manuscript before the end of rebuttal.
>
> **On the clarity of the description of the method**. We are sorry that the manuscript in submission are less clarified in the methodology design. In the current modified version, we have included a running example and the pseudo code in the appendix. Please comment on the discussion thread to shepherd us for revision.
>
> In the formal revised manuscript, we will include the formulas of the committee-voting. Indeed, Consensus uses a simple normalization-averaging procedure to obtain the aggregated interpretation result, which approximates to the ground truth tightly.
>
> We are regret for misleading on dimensionality issues. Actually, there are no dimensionality difference issues in RBF. Please refer to the pseudo code in the appendix of the current modified version for details.
>
> The significance tests were carried out using Spearman’s correlation to correlate the ranking lists, and Pearson’s correlation to correlate scores. Both use T-test to calculate the p-value based on the correlation coefficients. (Please respond us asap if we did anything wrong here. We could prepare a fix.) For the formulas for the correlation scores, we follow the standard implementation in SciPy.  For the final aggregating scores, please refer to the pseudo code in appendix of the current modified manuscript for details. To address your comments, we will include all these descriptions in the texts in the formal revised manuscript.
>
> For the tables reporting every CNN’s score, please refer to Table 1 in appendix of the current modified version, where we present the accuracy, interpretability scores using both Consensus (LIME) and Consensus (SmoothGrad) for more than 80 models used in the experiments.

---

> ### Author Response · Authors · 2020-11-18
> **Response to ICLR 2021 Conference Paper1036 AnonReviewer4 (Part 2/2)**
>
> **On Experimental comparisons with other methods**. Sorry for misleading. The purpose of Consensus is to evaluate the interpretability of DNN models for visual object classification tasks. In this way, Consensus adopts LIME and SmoothGrad for obtaining the interpretation results. We don’t attempt to test Consensus on the tasks other than the visual object classification. LIME and SmoothGrad would not be able to appropriately respond to the features of scenes, textures, or materials, when the models were trained for visual object classification. To address your concerns, we will revise the manuscript accordingly, to highlight the applicability of proposed methods.
>
> Sorry for misleading. Actually, models could be fine-tuned from pre-trained one. Indeed, in our work, only the experiment based on ImageNet works on the models that were trained from scratch in an end-to-end manner. The rest experiments for CUB-200-2011, Cars-196, Oxford Flowers 102, and Foods 101, use the DNNs fine-tuned from the pre-trained models (not from scratch). To address your comments, we will revise the manuscript accordingly.
>
> The key idea of our paper is that through aggregating the explanations of a (relatively small) number of networks, which even were randomly drawn from a pool of networks, could figure out or approximate to the ground truth of human-readable explanation. With such approximated or quasi ground truth, one can freely scale-up the overall evaluation of interpretability of deep models on large datasets, even when the human annotated ground truth is not available. Actually, in the experiment of Figure 7, we compare the Consensus’s aggregated interpretation results with the ground truth, and we repeat the experiments on every committee size for 20 times independently. The randomized independent trials clearly showed that even when the committee size is relatively small (e.g., 15), Consensus’s approximation to the ground truth is tight. With the increasing committee size, the variance of evaluation results is marginally decreased. The results clearly stated that randomly-picked networks are effective and robust enough to approximate to the ground truth, when they vote as a committee with equal weights. That is what we call “democratizing the evaluation of interpretability” in the title of manuscript. The DNN models contain some instinct sense on interpretability, which allows them to do the peer evaluation on interpretability. To address your concerns, we will discuss the key contribution of the paper and highlight the discussion in the revised manuscript.
>
> Thanks for the suggestion on the use of MS-COCO / LVIS dataset. Actually, in the current modified manuscript, we already include a visual example in the comparison among the interpretation results of different networks based on MS-COCO images using both LIME and SmoothGrad, the Consensus’s aggregated interpretation result, and the ground truth of the visual segmentation. The comparison clearly states that (1) Consensus’s aggregated interpretation approximates to the ground truth tightly, and (2) Consensus’s aggregated interpretation is much better than any single network. The visual comparisons backup our intuition of method design. To address your concerns, we will include more visual examples in the formal revised manuscript.
>
> **On Style / visuals**. We are sorry for the errors in the language and visualization. We have carefully followed your instruction to fix the errors. Thanks!
>
> Many thanks for your comments. We have included a running example, a list of pseudo codes, and several visual examples based on MS-COCO datapoints in appendix of the current modified manuscript. We are working hard on revising the manuscript to address all your comments, especially in the clarification of the algorithm design and computation. Your comments are valuable to us. Please feel free to comment on the discussion thread and shepherd us for improving the manuscript. We will upload the formal revised manuscript before the end of rebuttal. Thanks!

---

> ### Author Response · Authors · 2020-11-24
> **Any further comments?**
>
> Dear Reviewer,
>
> We are going to offer the formal revision by the end of rebuttal period. We hope our previous responses and current revised manuscript could address your concerns. Do you have any futher comments that we could follow to make the final revision? Many thanks for your attentions.
>
> Cheers,
>
> Authors

---

### Official Review · AnonReviewer2 · 2020-10-27
**Some aspects need to be addressed**

**Rating:** 5
**Confidence:** 5

**Review:**

 The manuscript proposes "Consensus" an evaluation method to measure the interpretability of a given deep model without the need of a dataset with annotated ground-truth concepts.

 The proposed method works by leveraging a set of pre-trained deep models (committee), and a reference model explanation method. The explanation method is use to produce explanations for every image/sample on a target dataset, and the aggregated explanation from every model in the committee is considered as "approximated/quasi ground-truth". The interpretability of a given model is defined as the mean similarity between all the explanations produced by the model in question and the quasi ground-truth (aggregated explanation) produced by the committee.

Strong points
- Experiments cover a good amount of deep models.
- An ablation study is in place.

Weak points
- Terminology is ambiguous.
- limited number of model explanation methods.

On the stronger side, the evaluation of the proposed method is relatively good, it includes an ablation study considering several relevant factors of the proposed method, e.g. committee size, and the effect of the members that are part of the committee.
Moreover, it validates the proposed method considering a large amount of deep models.

On the weaker size, there are several aspects that I feel need to be addressed:

I had some difficulties while reading the manuscript. The use of specific terminology is quite ambiguous at times. In several parts of the manuscript there are terms like "variety of interpretability of models", "interpretation result", "interpretability evaluation" that are not standard and hinder the content to a good extent. I would suggest very early on the manuscript defining these terms so that their meaning is clear much later in the body of the paper.

Very related to the previous point, the tasks of model interpretation (identifying the information/features encoded internally in a given model) and model explanation (justifying the predictions made by a given model on a given sample) seem to be used interchangeably. In my opinion, this is another strong source of confusion.
This should be addressed when discussing methods like LIME and SmoothGrad which are model explanation methods.
Moreover, this seems to actually point to the definition of one of the ambiguous terms mentioned in the previous point. It seems that "interpretation result"  actually refers to the explanation produced by a given sample.

The main characteristic of the proposed method is its independence on annotated ground-truth concepts. However, as admitted by the manuscript, there is still the requirement of having the data needed to train the models in the committee. This still sounds like a expensive requirement.
In addition, at this point it is not clear how the proposed consensus method would apply to a task/application (e.g. medical imaging) where data is scarce and deep models are not available in abundance. Discussing this direction would strengthen the manuscript.

In the comparison against the Network Dissection method (Sec. 3.3), it is stated that the rankings produced by the proposed method are quite similar to those of Network Dissection, but with differences between the ResNet152 and DenseNet161 models. Is it possible to get an insight on the source of this difference? Is it possible to know why the proposed method prefers one over the other?

When assessing the effect of the size of the committee (Sec. 6), what if just few specific models are responsible for the observed performance? then the number of randomly selected models might not be so relevant as the selection of these few models. Perhaps an incremental approach when models are gradually integrated in the committee should be analysed.
This setting, of few models determining the performance, seems to be suggested by the change in mean and standard deviation on performance in Fig.7. For committees with few models (4-9) the large standard deviation pushes performance (of some random committees) closer to the upper bound (0.70). The consideration of committees with larger sizes, and thus the likely potential inclusion of these relevant models, just makes the mean performance converge faster.

---

> ### Author Response · Authors · 2020-11-18
> **Response to ICLR 2021 Conference Paper1036 AnonReviewer2 (Part 1/2)**
>
> Many thanks for your review and the constructive comments. We do appreciate your efforts in helping us to improve the manuscript. Actually, we have uploaded a modified version with some updates in appendix. We will update the formal revised version of the manuscript before the end of rebuttal.
>
> We do appreciate your efforts in helping us to improve the manuscript. Actually, in our paper, we don’t explicitly differentiate the concepts of interpretation and explanation. We are regret for the interchangeable use of these two terms. Accordingly to the definitions you mentioned in the comment, Consensus is based on the settings of explanation “(justifying the predictions made by a given model on a given sample)”. Moreover, Consensus aims at approximating the overall quantification, here namely interpretability, that compares the explanation of the model (obtained by LIME or SmoothGrad) with human-readable explanations on a large dataset, when the ground-truth of explanations is not available. To address your comments, we will revise the manuscript accordingly. Actually, in the current version of modified manuscript, we already provide a running example, which clarifies the terminologies with the samples, in the appendix. In the formally revised manuscript, we will include these examples and clarification in the introduction section.
>
> Many thanks for your comments on the cost of our proposed methods. Actually, we have tried to way to reduce the cost of training networks for the committee. In our experiments, while the networks for the experiments based on ImageNet were trained from scratch in an end-to-end manner, rest experiments all use the networks fine-tuned from pre-trained models, where few training epochs are enough for the low-cost transfer learning. We are very sorry that we haven’t tried the models and datasets for the applications/tasks like medical imaging. It is really a good direction for future work. To address your concerns, we will include the cost issue of Consensus and the application domain in the discussion of the formal revised manuscript.
>
> Thanks for your insightful comments on the comparisons between the ranking results of Consensus and Network dissection. Actually, we believe two results are close enough from the perspectives of ranking lists. Only one pair of networks in 5 networks (20 pairs) are with conflict orders. Indeed, Consensus and Network Dissection evaluate the interpretability of the networks in different ways. While Consensus matches the explanation with the visual objects on images (due to the results of LIME and SmoothGrad), Network Dissection counts the number of neurons responding to the visual objects and patterns (color, materials, textures, scenes, and parts). Furthermore, Network Dissection evaluates the interpretability of the DNN classifiers trained on ImageNet using the Broden dataset, while our experiment here doesn’t assume the needs of additional dataset other than the validation set of ImageNet. In this way, the results by Consensus and Network Dissection might be slightly different. To address your concerns, we will include the discussion on the comparison of these two evaluation frameworks in the formal revised version.

---

> ### Author Response · Authors · 2020-11-18
> **Response to ICLR 2021 Conference Paper1036 AnonReviewer2 (Part 2/2)**
>
> We do agree with you that the contribution of some models to the “correctness” of the committee-voting might be different. However, we don’t intend to get a conclusion that “few specific models are responsible for the observed performance”, no matter for good or bad performance demonstration. Indeed, the key idea of our paper is that through aggregating the explanations of a (relatively small) number of networks, which even were randomly drawn from a pool of 80+ networks, could figure out or approximate to the ground truth of human-readable explanation. Actually, in the experiment of Figure 7, we compare the Consensus’s aggregated interpretation results with the ground truth, and we repeat the experiments on every committee size for 20 times independently. The randomized independent trials clearly showed that even when the committee size is relatively small (e.g., 15), Consensus’s approximation to the ground truth is tight. With the increasing committee size, the variance of evaluation results is marginally decreased. The results clearly stated that randomly-picked networks are effective and robust enough to approximate to the ground truth, when they vote as a committee with equal weights. That is what we call “democratizing the evaluation of interpretability” in the title of manuscript. Of-course, searching a subset of dominating networks might improve the efficiency of Consensus. However, the selection might be data driven and additional efforts is required in a case-by-case manner.
>
> Again, we do appreciate your review and constructive comments. We are encouraged to further revise the manuscript. Before the end of rebuttal, we will upload a formal revised version of the manuscript with all comments well addressed. Please feel free to comment on the thread of discussion and shepherd us for improving the manuscript.

---

> > ### Comment · AnonReviewer2 · 2020-11-22
> > **RE: Response to ICLR 2021 Conference Paper1036 AnonReviewer2 (Part 2/2)**
> >
> > Thanks for addressing the points stressed in my review.
> >
> > I will be looking forward to the revised version that you plan to upload prior the end of the rebuttal period.
> >
> > stay safe

---

> > > ### Author Response · Authors · 2020-11-24
> > > **Many thanks for your encouraging comments**
> > >
> > > Many thanks for your encouraging comments. We are going to offer the formal revision soon. Hopefully, the revised one would address all your concerns. Hope the revised manucript could receive your full consideration for acceptance.

---

### Official Review · AnonReviewer3 · 2020-10-28
**The paper presents an interesting approach for unlabelled evaluation of ML explainers, but (i) since the contribution is largely based on the extensive evaluation, the paper needs several improvements in terms of design choices and (ii) the usability of the proposed method (e.g. in terms of robustness evaluation of the model or quality of the explainer) should be better clarified.**

**Rating:** 4
**Confidence:** 4

**Review:**

The paper deals with explainable machine learning in the supervised setting and especially tackles the case where no ground truth data for evaluating the generate explanations, such as bounding boxes for objects, is available. The proposed "Concensus" approach retrains established architectures on the target dataset and averages their generated explanations from out-of-the-box explainers, such as LIME or SmoothGrad. The approach is evaluated for image classification on ImageNet and CUB-200-2011 by comparing the averaged explanations of the committee models when using LIME and SmoothGrad with the ground truth bounding box and segmentation, respectively. Minor evaluations are included for datasets Stanford Cars 196, Oxford Flowers 102  and Foods 101. The results show that the averaged explanations strongly correlate with the mean average precision with respect to the label distances.

The approach tackles the interesting setting of evaluating explainable ML approaches - here without labelled objects for computer vison tasks. In general, I find it interesting to evaluate such kind of ensemble-based approach for generated explanations, which is a common technique for adding model robustness or approximating uncertainty, and is closely related to bootstrap aggregation. To this end, the authors provide a quite large variety of experiments for different established vision benchmark datasets, including ablation studies on varying the committee size or randomly generated committees. However, I am left with several open questions for the approach, which I summarize below. In general, however, I am missing the added value of the approach with respect to the evaluation of explainable ML, as it remains unclear to me (based on the approach description and conducted evaluation) if the proposed tool can actually replace empirical evaluations in research or, even more important (and possibly intended by the authors) could support end-users in evaluating their models via out-of-the-box explainers.

I am wondering why the networks always need to be trained from scratch for a new task/dataset? Wouldn't the approach work for standard transfer learning? Here, it would be interesting to explore smaller, challenging datasets as well.

Also, why is the use of "known models" for buliding the committee emphasized? I am aware of prior works which argue that activation-based explainers are bias from the underlying architecture, but the training procedure should have significant impact on the "explanability" (at least as measured by approaches such as SmoothGrad) as well. Wouldn't it make sense to vary the training (or additional fine-tuning) parameters/used augmentation strategies as well?

As the paper highlights the unlabelled evaluation setting, I am missing related work in the field of explainable / interpretable ML, especially on counterfactuals. Approaches such as [1,2] generate such counterfactual explanations by altering the image towards other, potentially risky classes. Could you comment on the relation of the proposed approach to these kinds of works?

For the evaluation, the authors use labelled datasets for empirically validating their claims. I am wondering - and this might be a general point for such an evaluation - why/how the explanation component can be evaluated independently of the model, as imperfect explanations might uncover model weaknesses. Is your evaluation affected by this? In addition, as mentioned before with respect to the applicability of Consensus to transfer learning, I am wondering if the evaluation should cover cases where the network might overfit or misperform. Would your approach potentially rule out possibly interesting misbehaviors of a network in such a case? Lastly, could you provide design choices for the evaluation parameters, such as committee size or number of sampled networks?

References:
[1] Goyal, Y., Wu, Z., Ernst, J., Batra, D., Parikh, D. and Lee, S., 2019, May. Counterfactual Visual Explanations. In International Conference on Machine Learning (pp. 2376-2384).
[2] Liu, S., Kailkhura, B., Loveland, D. and Han, Y., 2019, November. Generative Counterfactual Introspection for Explainable Deep Learning. In 2019 IEEE Global Conference on Signal and Information Processing (GlobalSIP) (pp. 1-5). IEEE.

+++ Updates after author response +++
I want to thank the authors for their answers as well as their attempts to improve the manuscript. I have read the other reviewers' comments and the updated version of the paper. The latter improves on the clarity of the approach in terms of the formal presentation of the approach, but a concise problem definition is still missing in my opinion.

I feel that the paper still needs to additionally quantify who many labels are saved compared to classic cross-validation, as the models still need to be trained from scratch or fine-tuned. Here, it is important to establish when the method is actually (guaranteed to be) sufficiently concise, such that it can be used in practice. I therefore keep my tendency to reject the current version of the paper.

---

> ### Author Response · Authors · 2020-11-18
> **Response to ICLR 2021 Conference Paper1036 AnonReviewer3 (Part 1/2)**
>
> Many thanks for your review and constructive comments. We do appreciate your efforts in helping us to improve the manuscript. Actually, we have uploaded a modified version with some updates in appendix. We will update the formal revised version of the manuscript before the end of rebuttal.
>
> We do believe the added value of the proposed approach is to scale-up the evaluation for the interpretability of DNN image classifiers when the ground truth of “visual objects” is not available. Actually, it is not difficult to use the out-of-box explainer (or interpretation tools such as LIME and SmoothGrad) to explain/interpret the behaviors of DNN classifiers on certain individual images and check the explanation/interpretation manually by human experts. However, at the scale of a large dataset, it is impossible for human expert to verify the explanation/interpretation results one-by-one, and further quantify the interpretability for the whole model in a low cost. In such a way, the key contribution of our work is to provide a systematic protocol to automate, and even scale-up the evaluation of interpretability of DNN classifiers on large datasets, when the ground truth of visual objects is not available. We are not intending to “replace empirical evaluations in the research”.  To address your comments, we will include the discussion on the applicability of proposed methods in the revised version of the manuscript.
>
> In terms of “training from scratch”, we are regret for misleading. We said the networks are trained from scratch in the methodology section. We do understand in practical deep learning, fine-tuning or transfer learning from pre-trained models is quite often. Indeed, many of our experiments, including CUB-200-2011, Cars-196, Oxford Flowers 102, and Foods 101, train the DNNs from the pre-trained models (not from scratch). On the other hand, we train DNN for ImageNet all from scratch. In this way, our experiments include both DNN models trained from scratch or fine-tuned from pre-trained models, while Consensus as an interpretability evaluation tool works well under both settings for all experiments/ablation studies. To address your comments, we will revise the algorithmic section accordingly and fix this issue.
>
> In terms of “known models”, we are sorry for misleading again. We believe it is a language issue. We emphasize the “known models”, so as to make difference between the “new model” that is the target for interpretability evaluation on a dataset. Indeed, we didn’t give any constraints on selection of models to build a committee. In our experiments, we include famous or widely-used models, such as ResNet, VGG, Inception, DenseNet, MobileNet families and their variants, and also include the models such as SqueezeNet, SuffleNet, and AutoDL (some searched architectures). Furthermore, in Figure 7 of the current version, we introduce an experiment that evaluate the interpretability using the committee with networks that are randomly picked up from a pool. Certain robustness and consistency has been demonstrated. In this way, the networks used to form committee might not be a significant issue for building the committees.

---

> ### Author Response · Authors · 2020-11-18
> **Response to ICLR 2021 Conference Paper1036 AnonReviewer3 (Part 2/2)**
>
> For the impact of training procedure, actually, we follow the default settings to train every DNN classifier. In our paper, we follow the settings used in [1] that evaluates and compares the interpretability of DNN classifiers caused by the use of different models. Thus, the evaluation and comparisons of interpretability of DNN classifiers of the same model but trained with different hyper-parameters/strategies are not considered here. But definitely, this is a promising direction of future work. To address your concerns, we will discuss this issue in the revised version of the manuscript.
>
> We do appreciate reviewers point out the important references [2] and [3]. Our work actually focuses on evaluating the interpretability of DNN classifiers when the ground truth for interpretation is not available. We propose to use LIME and SmoothGrad as two alternative algorithms to obtain the interpretations (for evaluation purposes). We agree with you that we also can adopt counterfactual based algorithms [2,3] to obtain interpretation of DNN models on certain samples, while reusing the whole idea of committee-voting for interpretability evaluation. To address your comments, we will include these two papers in the discussion of the revised manuscript.
>
> In terms of evaluation, we do agree that it is challenging to evaluate an “evaluation component” independently. We have done comprehensive experiments including more than 80+ networks to evaluate our algorithms. We include ablation studies to verify the functions of key components used for interpretability evaluation. Please see the appendix of current modified version (it is not the final revised manuscript), where we include Table 1 consisting of detail results of all these 80+ models. We have tried our best to avoid internal and external threats to validities. Of-course, we would like to discuss these threats in the revised manuscript.
>
> After all, the goal of interpretation is to understand how a model makes the prediction (e.g., predictors used and their importance), while the goal of interpretability evaluation is to quantify the difference between the interpretation results of a model and the ground-truth of visual objects labelled by human experts. In this way, Consensus aims at providing an overall metrics for the interpretability evaluation on a large dataset, rather than pointing out the imperfect interpretation or any “interesting misbehavior” of a model on individual cases. But we agree with you that the abnormal detection in the committee networks might help us to identify some “interesting misbehaviors” that is significantly different from the rest of committee members. Furthermore, the interpretability evaluation is not the performance evaluation. We don’t expect Consensus help us identify the models that are over-fitted. However, maybe evaluating and comparing the interpretability of over-fitted models with the well-trained one would provide some interesting insights. To address your comments, we will include the discussion in the revised manuscript.
>
> The experiments in Figure 7 presents the facts that Consensus is committee-size efficient. Using a small number (e.g., 15) of networks can already achieve similar results, compared to the evaluation based on a large number (e.g., 85) of networks. All networks are randomly picked up from a pool of commonly-used networks. Thus, Consensus is quite robust.
>
> Again, we do appreciate your review and constructive comments. We are encouraged to further revise the manuscript. Before the end of rebuttal, we will upload a formal revised version of the manuscript with all comments well addressed. Please feel free to comment on the thread of discussion and shepherd us for improving the manuscript.
>
> [1] Bau D, Zhou B, Khosla A, et al. Network dissection: Quantifying interpretability of deep visual representations[C]//Proceedings of the IEEE conference on computer vision and pattern recognition. 2017: 6541-6549.
>
> [2] Goyal, Y., Wu, Z., Ernst, J., Batra, D., Parikh, D. and Lee, S., 2019, May. Counterfactual Visual Explanations. In International Conference on Machine Learning (pp. 2376-2384).
>
> [3] Liu, S., Kailkhura, B., Loveland, D. and Han, Y., 2019, November. Generative Counterfactual Introspection for Explainable Deep Learning. In 2019 IEEE Global Conference on Signal and Information Processing (GlobalSIP) (pp. 1-5). IEEE.

---

> > ### Comment · AnonReviewer3 · 2020-11-23
> > **Response**
> >
> > Thank you for the clarifications! I believe the approach as great potential, but I still have the feeling the paper could be strengthened in terms of (i) formally introducing the approach and (ii)  evaluating the approach in terms of the actual added value for end-users.
> >
> > I agree with the point raised by other reviewers that the paper needs to quantify the "saved" work on labeling data for the evaluation, as the individual models need to be trained (either fine-tuned or trained from scratch). Even in the case of fine-tuning, it would be beneficial to quantify the number of labels needed with / without the method. Did you gather empirical insights here? For example, is the method (more) usable for small sample sizes compared against withholding few labeled samples for evaluating interpretability?

---

> > > ### Author Response · Authors · 2020-11-24
> > > **Re: Response**
> > >
> > > Thank you very much for your recognition to our work and the follow-up suggestions! We are improving our manuscript following the directions as you proposed: (i) formally introducing the approach through an illustration, a pseudocode and the improved description; (ii) discussing the added value of our proposed approach in different scenarios, including for end-users. Thanks for your sherpherd.
> > >
> > > We also agree with you and other reviewers about the potential usage of our proposed approach. Indeed, in Figure 2 (Ground-Truth-based v.s. Consensus-based Interpretability Evaluation), our experiment on CUB-200-2011, where the ground truth of interpretations is available, shows that the Consensus-based approach obtains the ranking of evaluated models that is significantly correlated to one obtained by the ground truth based approach. This shows that our method is equally usable against withholding labeled samples for evaluating interpretability, while our method does not rely on the ground truth.
> > >
> > > Furthermore, we would like to discuss a little bit more about the potential usage/applicability of our approach. Actually, our proposed approach with a small number of trained models could approximate the ground truth of interpretations for each data sample in the dataset (Please refer to Figure 7 and visual examples in Figures 10 and 11 in the current online version). For the interpretability evaluation of any models on any dataset, the ground truth based evaluation approaches rely on human subjective interpretations; instead, our approach can automate this process by just training a few more models and approximating the ground truth of interpretations. We thus say that the interpretability evaluation can be democratized without the need of ground truth as criterion through an electoral system. To highlight this part, we will include the discussion on these issues in the formal revised manuscript. Of-course, gaining more insights from both empirical and theortical aspects, including sample sizes and the other factors, might be the future direction of our research. We will discuss the future work in the revised version.
> > >
> > > We are making our best efforts to improve the quality of manuscript following your suggestions, which will be available soon. Please let us know if further improvements are needed. Thanks for your time and shepherding us for improving the manuscript.

---

### Official Review · AnonReviewer1 · 2020-10-28
**Official Blind Review**

**Rating:** 6
**Confidence:** 3

**Review:**

The paper seems to introduce a very important model to evaluate intepretatbility of neural networks.
However, the paper is not extremely clear and intepretable.
The main idea is to pool together interpretations coming from different systems and then selecting the best interpretation by voting. The procedure and the model is described in a single section, that is, section 2. It is extremely obscure the way Consensus operates. A running example can be very relevant in this section. Without this running example, it is difficult to attract readers into your paper.
For example, one between Figure 4 and Figure 5 could be this running example? Can you use it to explain your model?

---

> ### Author Response · Authors · 2020-11-18
> **Response to ICLR 2021 Conference Paper1036 AnonReviewer1**
>
> Many thanks for your review and encouraging comments. We do appreciate your efforts in helping us to improve the manuscript. Actually, we have uploaded a modified version with some updates in appendix. We will update the formal revised version of the manuscript before the end of rebuttal.
>
> In the current updated manuscript, we provide an early version of running example and the pseudo code in the appendix. Please have a look and give us comments to further improve the manuscript. In the formal revised version, we will include the running example in the introduction section and revise the algorithmic section accordingly to make it clearer.
>
> Many thanks for your comments. We will get back to you soon with the formal revised manuscript. Hope the revised version could receive your full consideration for the publication. Please feel free to comment on the thread of discussion and shepherd us for improving the manuscript.

---

### Author Response · Authors · 2020-11-24
**Summary**

We thank AC’s efforts in organizing the review. Many thanks to the reviewers for the recognition to our work and the valuable suggestions for improving the quality of our manuscript!

Following your instructions, we have uploaded a new version with suggested improvements and clarifications as mentioned in our responses during the discussions with reviewers. Here we summary the main modifications with respect to the initial submission.

Generally, reviewers raised up four major issues to our manuscript and we address these issues in the revised manuscript and our responses to reviewers.

1. Clarity of the methodologies

To address the concerns and make it clear, we include a running example (Figure 1 in Page 2, Section 1) and the pseudo code (Algorithm 1 in Page 4, Section 2) of the proposed framework. Furthermore, we also include the formulas to calculate the similarity between interpretations, and the formulas for voting (interpretation aggregation) in Section 2. A comprehensive version of Pseudo code has been given in the appendix (Algorithm 2 in Page 21, Appendix) to make it clearer. We also clearly introduce the metrics used for evaluation and others. Hopefully, we clarify the way we design and validate the Consensus now.

2. Contribution/Scope of our work

Reviewers also have concerns in the contributions we made in the manuscript from the perspective of added values and the scope of our work. For the controversies between "interpretation" vs. "explanation", in our manuscript, we use them interchangeably. We do understand in some context, they have different definitions. However, to make it clear, we visualized the way we define the interpretation (i.e., important predictors used by the model for classification on every data point) and interpretability (i.e., in which degree the interpretations of a model match the ground truth of interpretations) in the running example (see also in Figure 1 in Page 2, Section 1). For added values, we have discussed this issue in the Conclusion section (Page 9, Section 6). We emphasized that (1) when the human-labeled ground truth of interpretations is not available for interpretability evaluation, one could approximate the ground truth of interpretation through aggregating interpretation results of a small number of DNN classifiers; (2) with the "quasi-ground-truth", interpretability evaluation can be democratized through an electoral system constructed by the DNN models themselves, rather than the use of human labeled ground truth as criterion. To verify our two claims, we include (1) additional visual examples on the ground truth approximation (i.e., Consensus achievement in Appendix), and (2) more discussion the committee size and random committee formation (Figure 8 in Page 9, Section 5). Hopefully, the revised version addresses your concerns.

3. Experiment Details, including committee size, significance test, and scores

Per request, we include additional experiment results and elaborate the details on our experiments in both rebuttal responses and the revision. In Table 1 (in Page 22, Appendix) we include a numerical report of 81 (for ImageNet) + 85 (CUB-200-2011) models, including their validation accuracy, mAP between ground truth to the interpretations (using either LIME and SmoothGrad). We also include more details about the significance test, including p-value, Spearman’s correlation coefficients for correlating the ranks and Pearson’s correlation coefficients for correlating the values, and the discussion on the ranks of models (in Page 7, Section 3). We emphasize that only incorporating a small (random) subset of trained models as the committee, Consensus can approximate the ground truth well. Many thanks for your suggestions. Hopefully, the revised version addresses your concerns.

4. Additional experiments based on other datasets

Per request, we also include the experiments based on additional dataset, such as MS-COCO. Please see the visual example and comparisons on MS-COCO data in Figure 14 (Page 23, Appendix). Please note that, in addition to ImageNet and CUB-200-2011, we also carried out our experiments using Oxford Flower-102, Stanford Cars 196, and Foods-101. Actually, in current version of manuscript, we have clarified that the scope of our research is to evaluate the interpretability of DNN classifiers. We believe the validation of Consensus on the classification datasets is appropriate. Thanks for your suggestions!

Many thanks for your shepherding and guidance in these days. It has been a quite enjoyable period for us to improve the manuscript. We hope this paper could receive your full consideration for acceptance.

---

### Decision · Program_Chairs · 2021-01-07
**Final Decision**

**Decision:**

Reject

**Comment:**

The reviewers all found that the Consensus method introduced seemed sensible and applauded the authors on their extensive experiments.  However, clearly they struggled to understand the paper well and asked for a clearer and more formal definition of the methods introduced.  Unfortunately, the highest scoring review was also the shortest and also indicated issues with clarity.  It seems like the authors have gone a long way to improve the notation, organization and clarity of the paper, but ultimately the reviewers didn't think it was ready for acceptance.  Hopefully the feedback from the reviewers will help to improve the paper for a future submission.